# MTVCraft: Tokenizing 4D Motion for Arbitrary Character Animation

**Yanbo Ding**[1,2,4,*], **Xirui Hu**[2,3,*], **Zhizhi Guo**[2,‡], **Yan Zhang**[2], **Xinrui Wang**[2],
**Zhixiang He**[2], **Chi Zhang**[2], **Yali Wang**[1,5,†], **Xuelong Li**[2,†]

[1]Shenzhen Key Laboratory of Computer Vision and Pattern Recognition, Shenzhen Institutes
of Advanced Technology, Chinese Academy of Sciences, Shenzhen, China
[2]Institute of Artificial Intelligence (TeleAI), China Telecom, Beijing, China
[3]School of Computer Science and Technology, Xi'an Jiaotong University, Xi'an, China
[4]School of Artificial Intelligence, University of Chinese Academy of Sciences, Beijing, China
[5]Shanghai Artificial Intelligence Laboratory, Shanghai, China

`{yb.ding,yl.wang}@siat.ac.cn,`
`xiruihu@stu.xjtu.edu.cn,`
`guozz2@chinatelecom.cn`

## Abstract

Character image animation has rapidly advanced with the rise of digital humans. However, existing methods rely largely on 2D-rendered pose images for motion guidance, which limits generalization and discards essential 4D information for open-world animation. To address this, we propose MTVCraft (Motion Tokenization Video Crafter), the first framework that directly models raw 3D motion sequences (i.e., 4D motion) for character image animation. Specifically, we introduce 4DMoT (4D motion tokenizer) to quantize 3D motion sequences into 4D motion tokens. Compared to 2D-rendered pose images, 4D motion tokens offer more robust spatial-temporal cues and avoid strict pixel-level alignment between pose images and the character, enabling more flexible and disentangled control. Next, we introduce MV-DiT (Motion-aware Video DiT). By designing unique motion attention with 4D positional encodings, MV-DiT can effectively leverage motion tokens as 4D compact yet expressive context for character image animation in the complex 4D world. We implement MTVCraft on both CogVideoX-5B (small scale) and Wan-2.1-14B (large scale), demonstrating that our framework is easily scalable and can be applied to models of varying sizes. Experiments on the TikTok and Fashion benchmarks demonstrate our state-of-the-art performance. Moreover, powered by robust motion tokens, MTVCraft showcases unparalleled zero-shot generalization. It can animate arbitrary characters in full-body and half-body forms, and even non-human objects across diverse styles and scenarios. Hence, it marks a significant step forward in this field and opens a new direction for pose-guided video generation. Our project page is available at `https://github.com/DINGYANB/MTVCrafter`. A scaled version has been commercially deployed and is available at `https://telestudio.teleagi.cn/generatevideo/creativeWorkshop`.

## 1 Introduction

Character image animation (Chang et al., 2025; 2023b; Xu et al., 2025b; Men et al., 2024), which aims to synthesize videos of a reference character image driven by pose sequences estimated from an

---

* These authors contributed equally, and this work was done during internship at TeleAI.
† Corresponding Authors.
‡ Project Leader.

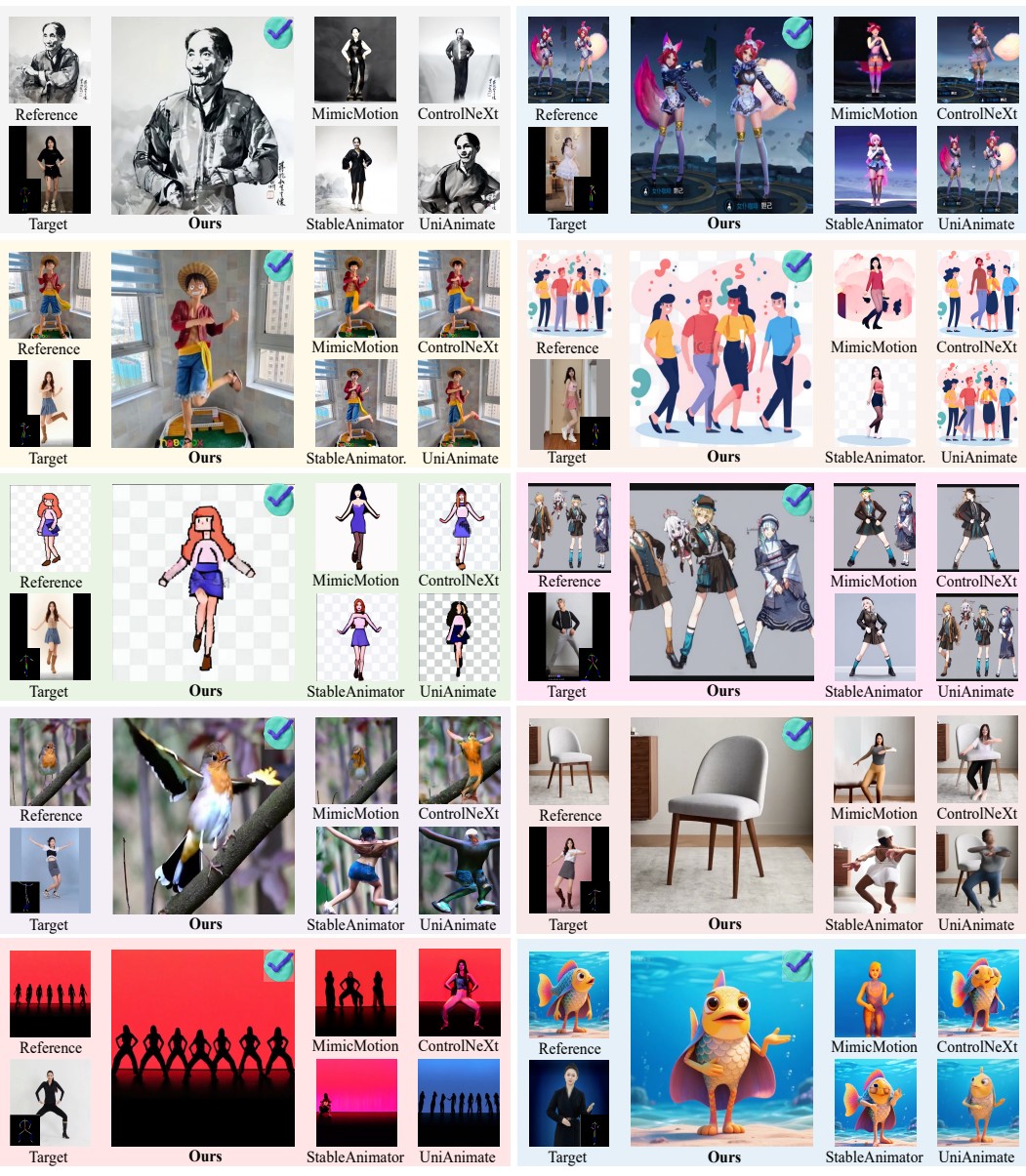

Figure 1: **Teaser.** We propose MTVCraft, a versatile framework that can effectively transfer pose sequences from a driven video in either full-body or half-body settings, while supporting a wide range of visual styles such as anime, pixel art, ink drawings, and photorealism. Beyond human characters, MTVCraft is further capable of handling non-human subjects such as animals and even inanimate objects, demonstrating superior robustness, strong generalizability to open-world scenarios, and the emergent ability to animate arbitrary characters.

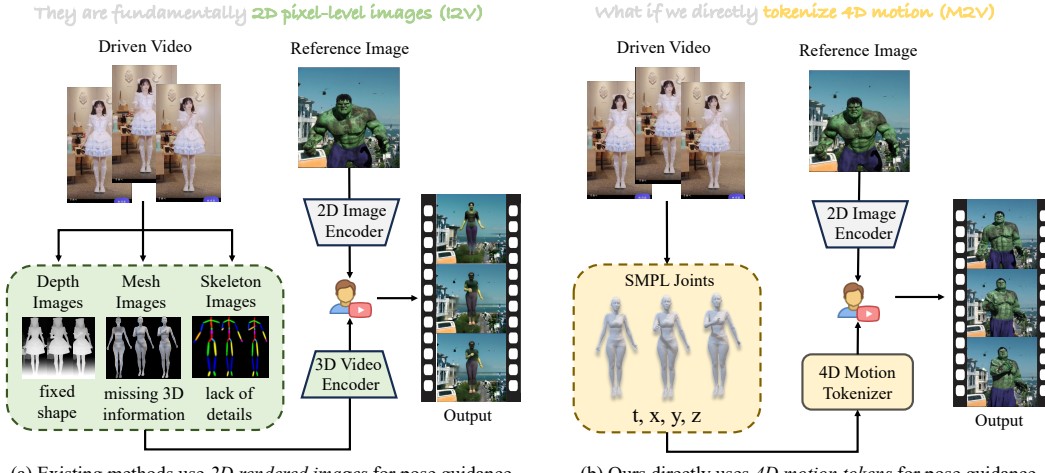

(a) Existing methods use *2D rendered images* for pose guidance.  (b) Ours directly uses *4D motion tokens* for pose guidance.

Figure 2: **Motivation.** Directly tokenizing 4D motion captures more faithful and expressive information than traditional 2D-rendered pose images derived from the driving video.

input video, has attracted increasing attention due to its broad applications in digital humans (Lauer-Schmaltz et al., 2024; Hu et al., 2025b), virtual try-on (Islam et al., 2024; Song et al., 2024), and immersive content creation (Chamola et al., 2024; Qin & Hui, 2023). To meet the growing demand, numerous methods (Hu, 2024; Hu et al., 2025a; Tu et al., 2024; Zhu et al., 2024b; Xu et al., 2024; Gan et al., 2025; Zhang et al., 2024a; Peng et al., 2024; Wang et al., 2025; Zhou et al., 2025) have been proposed to achieve high-quality animation with realistic motion and consistent appearance.

However, as shown in Figure 2, existing methods depend on 2D-rendered pose images to provide motion guidance for the generative model. This introduces two fundamental limitations. First, although pose images provide basic structural cues, they inevitably discard rich spatial-temporal motion from the real 4D world. Hence, they struggle to synthesize physically plausible and expressive motions, especially in complex 4D scenarios (e.g., Gymnast in Figure 13). Second, when the pose is provided in images, the model tends to blindly copy the fixed-shaped poses pixel-by-pixel without grasping the underlying motion semantics. Consequently, the animation often exhibits distortions or artifacts, especially when the pose images from the driving video significantly deviate from the reference appearance in shape or position (e.g., Hulk in Figure 2). Hence, a natural question arises: *can we directly model raw 4D motion rather than 2D-rendered pose images for animation?*

To answer this question, we draw inspiration from recent advances in motion generation (Hosseyni et al., 2025; Guo et al., 2024; Jiang et al., 2023; Zhang et al., 2023a), where SMPL body parameters (Plappert et al., 2016; Guo et al., 2022) are first quantized and subsequently used for motion generation. Built upon this insight, we propose **MTVCraft** (**M**otion **T**okenization **V**ideo Crafter), a novel framework that combines a 4D motion tokenizer with a motion-aware video Diffusion Transformer (Zhu et al., 2024a; Kong et al., 2024; Zheng et al., 2024) for arbitrary character image animation. Firstly, to leverage richer spatial-temporal information in the 4D world than what can be captured by 2D image renderings, we propose **4DMoT** (**4D Mo**tion **T**okenizer) to directly quantize 4D human motion data (i.e., 3D joint coordinates over time). The resulting motion tokens faithfully preserve the information of raw motion, effectively addressing the first limitation of lacking explicit 4D information. Secondly, we propose **MV-DiT** (**M**otion-aware **V**ideo DiT) for controllable animation. By integrating 4D motion attention into DiT blocks, our MV-DiT effectively leverages motion tokens as context for vision tokens. This design eliminates the need to render pose images and enables the model to better learn underlying motion semantics, thereby addressing the second limitation of pixel-level copying. To further improve spatial-temporal modeling, we incorporate unique 4D positional encodings (1D temporal + 3D spatial) into the motion attention. With this unified design, MTVCraft can be easily applied to different model scales, from CogVideoX-5B (Yang et al., 2024b) to 18B on Wan-2.1-14B (Wan et al., 2025). Leveraging 4D motion tokenization and motion attention, MTVCraft establishes a new paradigm for character animation and demonstrates versatility.

Our contributions are summarized as follows: (1) We introduce MTVCraft, the first pipeline that directly models raw 4D motion instead of 2D-rendered pose images for arbitrary character image animation. (2) We introduce 4DMoT, a novel motion tokenizer that encodes SMPL joint coordinates into 4D compact yet expressive tokens, providing more robust spatial-temporal guidance than 2D pose image representations. (3) We design MV-DiT, a motion-aware video DiT model equipped with unique 4D motion attention and 4D positional encodings, enabling animation effectively guided by 4D motion tokens. We implement two versions of MV-DiT, corresponding to small and large model scales. (4) MTVCraft achieves state-of-the-art performance on the TikTok (Jafarian & Park, 2021) and Fashion (Zablotskaia et al., 2019) benchmarks. Moreover, as shown in Figure 1, MTVCraft showcases powerful zero-shot generalization to unseen motions, styles, scenarios, and characters, including full-body or half-body, and even non-human objects.

## 2 RELATED WORK

**Diffusion Models for Controllable Generation**  Diffusion Models (Sohl-Dickstein et al., 2015; Ho et al., 2020; Song et al., 2020) have achieved remarkable success in visual generation. Unlike traditional GAN-based methods (Lee & Seok, 2019; Xu et al., 2018; Goodfellow et al., 2020) which often suffer from training instability and mode collapse, diffusion-based approaches offer more stable training dynamics and can generate high-quality content with improved diversity. This superior performance has led Stable Diffusion series (Rombach et al., 2022; Esser et al., 2024; Podell et al., 2023) to quickly dominate the field of vision-generative AI. Some other works (An et al., 2026; Liang et al., 2026) systematically survey recent advances. To enable finer control beyond text, ControlNet (Zhang et al., 2023b) uses zero-initialization and network duplication to guide structural elements such as sketches and depth maps. ControlNeXt (Peng et al., 2024) improves this design with a lightweight module and cross-normalization strategy. AnimateDiff (Guo et al., 2023) introduces motion control by injecting temporal layers into text-to-video diffusion models. Other specialized methods extend controllability to aspects such as motion trajectories (Zhang et al., 2024b; Xiao et al., 2024), camera viewpoints (Hou et al., 2024; He et al., 2024), and scene layouts (Ding et al., 2024; Feng et al., 2023). In this work, we address the challenging task of arbitrary character animation, with the goal of achieving precise pose control across diverse characters in the 4D world.

**Character Image Animation**  Early approaches (Huang et al., 2021; Siarohin et al., 2021; 2019; Li et al., 2019) predominantly adopt GANs to animate the reference image, but struggled with visual artifacts. Recent advances in diffusion models have inspired their application to character image animation (Hu, 2024; Luo et al., 2025; Zhou et al., 2025). Disco (Wang et al., 2024a) first introduces a hybrid diffusion architecture with disentangled control over human foreground, background, and pose. MagicAnimate (Xu et al., 2024) and AnimateAnyone (Hu, 2024) developed specialized reference and pose networks to control appearance and motion, respectively. Champ (Zhu et al., 2024b) leverages mesh renderings for enhanced controllability, while Unianimate (Wang et al., 2024b) and Unianimate-DiT (Wang et al., 2025) integrates Mamba (Gu & Dao, 2023) into diffusion models for improved efficiency. MimicMotion (Zhang et al., 2024a) and Realiscance (Zhou et al., 2024) implement regional loss functions to mitigate distortion. StableAnimator (Tu et al., 2024) uses HJB-based (Peng, 1992; Bardi et al., 1997) optimization to enhance identity preservation. AnimateX (Tan et al., 2024) and AnimateAnyone-2 (Hu et al., 2025a) extend motion transfer to non-human subjects and environmental interactions, respectively. Human-DiT (Gan et al., 2025) and HyperMotion (Xu et al., 2025a) use DiTs to enhance the animation quality and temporal coherence.. Importantly, all these methods rely on 2D images for pose guidance, including skeletons (e.g. DWPose (Yang et al., 2023) and OpenPose (Cao et al., 2019)), SMPL (Pavlakos et al., 2019; Loper et al., 2023) renderings, or depth maps (Yang et al., 2024a)). Our work directly encode 4D joint coordinates without intermediate rendering, and designs motion attention in DiTs to effectively leverage 4D motion tokens.

## 3 METHOD

**Diffusion Transformer**  The Diffusion Transformer (DiT) (Kong et al., 2024; Zheng et al., 2024; Yang et al., 2024b), as a prevailing approach, integrates a Transformer-based backbone into the diffusion process. Using Patchify (Peebles & Xie, 2023) and Rotary Positional Encoding (RoPE) (Su et al., 2024), the denoising network can effectively process inputs with varying spatial and temporal

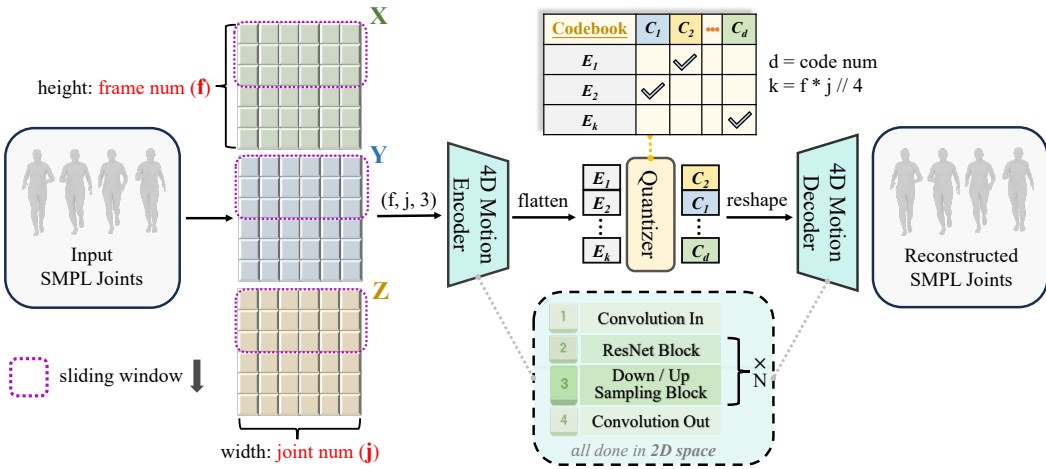

Figure 3: **Architecture of 4DMoT.** An encoder-decoder framework learns spatial-temporal latent representations of SMPL joint coordinates, and a vector quantizer learns 4D compact yet expressive tokens in a unified space. All operations are in 2D space along the frame and joint axes.

dimensions, thus improving scalability and adaptability compared with U-Net (Ronneberger et al., 2015). In practice, RoPE encodes relative positional information via rotation in complex space:

$$R_i(x, m) = \begin{bmatrix} \cos(m\theta_i) & -\sin(m\theta_i) \\ \sin(m\theta_i) & \cos(m\theta_i) \end{bmatrix} \begin{bmatrix} x_{2i} \\ x_{2i+1} \end{bmatrix} \tag{1}$$

where $x$ is the input query or key vectors, $m$ is the positional index, $i$ is the feature dimensional index. $\theta_i$ is the frequency, i.e., $10000^{-2i/D}$, and $D$ is the dimension of the attention layer.

**Overview** After the preliminary introduction, we next explain our MTVCraft in detail. In Section 3.1, we introduce 4DMoT for motion tokenization, where the resulting 4D motion tokens provide more robust spatial-temporal cues than 2D-rendered pose images. In Section 3.2, we present MV-DiT to animate characters conditioned on 4D motion tokens, featuring motion attention with unique positional encodings and motion-aware classifier-free guidance (Ho & Salimans, 2022). Finally, in Section 3.3, we show that our method is easily scalable to larger model sizes with only minor architectural adjustments, demonstrating its flexibility for practical deployment.

### 3.1 4D MOTION TOKENIZER

To guide character animation with rich 4D signals, we extract SMPL (Loper et al., 2023) sequences from the driving video as conditions. Prior works (Zhou et al., 2024; Pang et al., 2024; Zhu et al., 2024b) render 3D meshes into 2D images, often yielding deficient motion representation and introducing shape and position biases (Figure 2). In contrast, we directly tokenize raw SMPL sequences into 4D motion tokens, decoupling motion from absolute position and shape variations to obtain a compact, robust representation. We first curate a SMPL motion-video training dataset, then design a 4D motion VQVAE (Figure 3) to learn noise-free tokens for subsequent animation.

**Motion-Video Dataset Preparation** Existing open-source datasets like (Jafarian & Park, 2021; Zablotskaia et al., 2019), are limited in both motion diversity and visual quality, which constrains their effectiveness in training powerful generative models. To this end, we curated a diverse dataset comprising 200K video clips. These clips are sourced from public datasets and web-crawled content videos, covering a wide variety of human figures. The clips are then rigorously filtered to ensure high-quality (see Appendix B). For each remaining clip, we use NLF-Pose (Sárándi & Pons-Moll, 2024) to estimate SMPL joint rotations $\{\theta_t\}_{t=1}^T$ and root translations $\{\tau_t\}_{t=1}^T$, where $T$ is the number of frames. The estimated $\theta_t$ are combined with a standard human SMPL shape to compute 3D joint coordinates $J_t \in \mathbb{R}^{24 \times 3}$ using forward kinematics (Loper et al., 2023). Here, we use a standard neutral SMPL shape instead of the per-frame predicted one to decouple motion from individual

shape variations. Furthermore, the subsequent tokenization enhances this decoupling by learning motion representations independently of shape, allowing the model to focus on motion dynamics. The root translations $\tau_t$ are then added to $J_t$ to preserve position changes over time. The resulting $J_t$ serves as input to 4DMoT. The final training dataset comprises 30K high-quality SMPL motion-video pairs, averaging 600 frames per video and covering diverse motions and scenarios.

**Model Architecture of our 4DMoT**  Since the VQVAE is widely used for discrete tokenization in downstream tasks (Guo et al., 2024; Wang et al., 2024c; Ma et al., 2025), we build upon its architecture. As shown in Figure 3, our 4DMoT consists of an encoder-decoder for reconstructing 4D joint coordinates, and a quantizer for learning discrete motion tokens. The encoder-decoder captures spatial-temporal information, while the quantizer helps to remove noise and learn a compact representation in a unified space. Specifically, given a motion sequence $\{J_1, J_2, ..., J_f\}$ with $f$ frames and $j$ joints, we first normalize it using global statistical mean and standard deviation of the dataset, and then convert it into a relative representation $M$ by subtracting the first frame, so that the first frame has all-zero joint coordinates. All subsequent processes are based on these differential joint coordinates. In this way, we enable the model to learn relative motion patterns and thereby decouple motion information from absolute positions. This approach leads to more robust and disentangled spatial-temporal representations compared to traditional mesh renderings. The encoder then maps it into a continuous latent space via residual blocks with 2D convolutions along temporal ($f$) and spatial ($j$) dimensions, and downsampling blocks with average pooling. This design enables effective interactions across both time and joint dimensions simultaneously and yields latent representations $\{E_m \in \mathbb{R}^d\}_{m=1}^{(1+(f-1)//4) \times j}$ (first frame not downsampled, $d$ denotes the token dimension). Next, a vector quantizer discretizes $E$ via nearest-neighbor lookup in a learnable codebook $\{C_n \in \mathbb{R}^d\}_{n=1}^s$, where $s$ denotes the codebook size. The codebook is optimized with Exponential Moving Average (EMA) (Polyak & Juditsky, 1992) and codebook resetting technique (Hosseyni et al., 2025) to maintain usage diversity. Finally, the decoder, which has a similar structure to the encoder but with upsampling blocks, reconstructs the differential motion $\hat{M}$ from the quantized codes $C$. Moreover, to better capture long-range dependencies, we incorporate dilated convolutions and a sliding window strategy. The complete training objective combines a reconstruction loss with a commitment loss to ensure faithful reconstruction and codebook effectiveness:

$$\mathcal{L}_{\mathbf{vq}} = \underbrace{\|M - \hat{M}\|_1}_{\text{reconstruction}} + \beta \underbrace{\|E - \text{sg}[C]\|_2^2}_{\text{commitment}} \qquad (2)$$

Where $\text{sg}[\cdot]$ denotes the stop-gradient operation, $\beta$ is a hyperparameter to control the weight of the commitment loss, $E$ and $C$ are the latents before and after quantization, respectively.

**Tokenization Strategy: Coordinates vs. SMPL Parameters**  Unlike prior motion generation works that tokenize SMPL parameters (e.g., (Guo et al., 2024)), we opt to tokenize SMPL joint coordinates for two reasons. First, our task focuses on video generation rather than predicting SMPL parameters. Joint coordinates capture spatial continuity and provide explicit positional information directly aligned with pixel-level generation, making them better suited than SMPL parameters that encode rotations indirectly. Joint coordinates can also be naturally expressed in a differential form relative to the first frame, allowing the model to better learn motion dynamics while decoupling them from absolute positions. Second, tokenizing coordinates avoids potential instabilities and ambiguities inherent in SMPL rotation representations (e.g., axis-angle discontinuities). While prior works explored SMPL-parameter tokenization for motion generation, directly using such representations for video generation may not provide interpretable control. In practice, differential joint coordinates provide a more stable and robust representation, facilitating effective learning in 4D space.

## 3.2 4D MOTION VIDEO DIFFUSION TRANSFORMER

With compact and noise-free 4D motion tokens obtained from 4DMoT, we next leverage them to drive character image animation in MV-DiT. In this section, we describe how these motion tokens are incorporated as conditioning signals, including identity preservation, 4D positional encodings, 4D motion attention, motion-aware classifier-free guidance, and scaling to larger model sizes.

**Identity Preservation**  Maintaining visual consistency of the reference character image is critical for controllable animation. A common strategy in previous methods (Chang et al., 2023b;

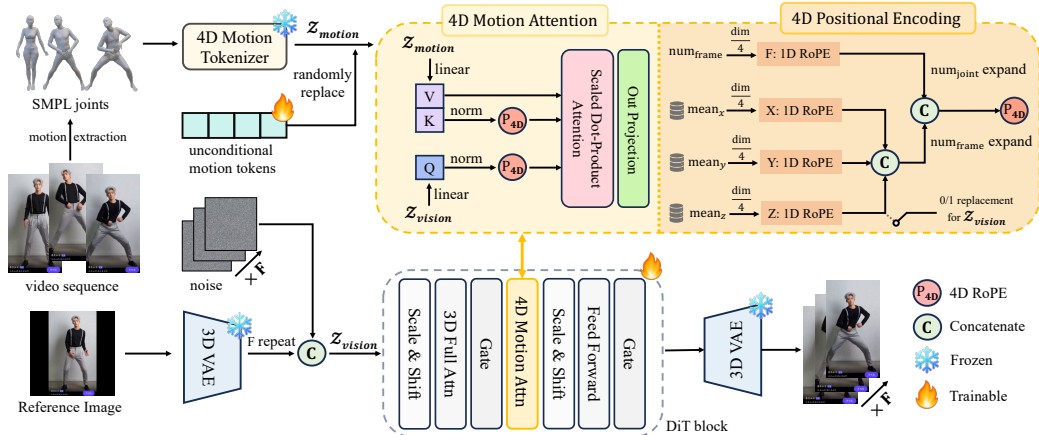

Figure 4: **Architecture of MTVCraft-6B.** Based on the video DiT model, we design unique 4D motion attention to leverage 4D motion tokens as context for vision generation. To enhance spatial-temporal relationships, we apply 4D RoPE over (t, x, y, z) coordinates to attention keys and queries.

Zhou et al., 2024; Chang et al., 2023a; Hu, 2024; Xu et al., 2024) is to introduce a separate reference network to model identity appearance independently. While effective, such designs increase architectural complexity and computational overhead. In contrast, we adopt a more straightforward yet effective repeat-and-concatenate scheme. Specifically, given the noisy video latents $\{z_t\}_{t=0}^{f} \in \mathbb{R}^{f \times c \times h \times w}$ and the reference image latent $z_{\text{ref}} \in \mathbb{R}^{c \times h \times w}$ obtained from a frozen shared VAE encoder, we compute the composite vision latents following:

$$z_{\text{vision}} = \text{Concat}\left(z_0, \text{Repeat}(z_{\text{ref}}, f)\right) \in \mathbb{R}^{f \times 2c \times h \times w} \tag{3}$$

This design explicitly injects identity information at every frame. Thanks to the 3D full self-attention mechanism in DiT, the model can directly establish interactions between video latents and reference image latents across spatial-temporal dimensions. As a result, identity consistency is preserved without introducing additional reference branches, leading to a simpler and more unified architecture.

**4D Positional Encodings**    To further enhance spatial-temporal relationships and enable more meaningful interaction between motion and vision tokens, we extend the standard 3D RoPE of vision tokens into 4D space and introduce dedicated 4D RoPE for motion tokens.

$$
\begin{aligned}
\text{Standard:} \quad & P_{\text{3D}} = \text{Concat}(R_t, R_h, R_w) \\
\Rightarrow \quad & P_{\text{4D}} = \text{Concat}(R_t, R_x, R_y, R_z)
\end{aligned} \tag{4}
$$

where each $R_*$ implements 1D rotary embeddings (Su et al., 2024) along a specific coordinate axis and is broadcast across other dimensions. The motivation for extending to 4D is two-fold. First, motion tokens naturally reside in structured 3D space evolving over time. Second, a unified positional formulation allows motion and vision tokens to share compatible geometric semantics during attention. As shown in Figure 4, for motion tokens $z_{\text{motion}}$, we compute positional encodings based on coordinates $(t, x, y, z)$, where $t$ denotes the frame index, and $(x, y, z)$ are the mean joint positions averaged over all frames, representing 24 joints in 3D space. Using dataset-averaged positions provides a unified reference for different human poses, offering stable and consistent positional cues that facilitate faster training convergence. Meanwhile, for vision tokens $z_{\text{vision}}$, which lack a depth axis, we use $(t, h, w)$ and assign $z = 0$ for the rotary embeddings along the depth dimension. This preserves the original 3D RoPE behavior while ensuring positional compatibility with motion tokens. With this 4D formulation, both motion and vision tokens can interact coherently within subsequent attention layers. Further implementation details are provided in Appendix F.

**4D Motion Attention**    To effectively leverage motion tokens $z_{\text{motion}}$ as context for vision tokens $z_{\text{vision}}$, we introduce 4D motion attention as shown in Figure 4, where $z_{\text{vision}}$ serve as queries and

$z_{\text{motion}}$ serve as keys and values, enabling the model to dynamically retrieve motion cues when generating spatial-temporal video representations. The attention mechanism is formulated as follows:

$$\mathbf{Q} = \text{RoPE}(\text{LayerNorm}(W_q(z_{\text{vision}})), P_{\text{4D}}^{\text{vision}}), \tag{5}$$

$$\mathbf{K} = \text{RoPE}(\text{LayerNorm}(W_k(z_{\text{motion}})), P_{\text{4D}}^{\text{motion}}), \tag{6}$$

$$\mathbf{V} = \text{LayerNorm}(W_v(z_{\text{motion}})), \tag{7}$$

$$\text{Attention}(\mathbf{Q}, \mathbf{K}, \mathbf{V}) = \text{Softmax}\left(\frac{\mathbf{Q}\mathbf{K}^T}{\sqrt{d_k}}\right)\mathbf{V}. \tag{8}$$

Here, $W_q$, $W_k$, and $W_v \in \mathbb{R}^{d \times d}$ are learnable projection matrices, and $P_{\text{4D}}^{\text{vision}}$ and $P_{\text{4D}}^{\text{motion}}$ denote the 4D RoPE applied to vision and motion tokens, respectively. The RoPE formulation follows Equation 1. The attention output is added to $z_{\text{vision}}$ via a residual connection, enabling motion-aware modulation while preserving spatial-temporal consistency of the video latents.

**Motion-aware Classifier-free Guidance**   To further enhance generation quality and generalization, we extend classifier-free guidance (CFG) to motion tokens. Standard CFG interpolates between conditional and unconditional predictions as $\hat{\epsilon}_\theta = \epsilon_\theta(z_t, t, c_\varnothing) + w(\epsilon_\theta(z_t, t, c_t) - \epsilon_\theta(z_t, t, c_\varnothing))$, where $\epsilon_\theta$ is the denoising network, $z_t$ is the noisy latent at timestep $t$, $c_t$ is the condition, and $w$ controls the guidance strength. However, motion tokens lack a natural unconditional form. We therefore introduce learnable unconditional motion tokens $c_{mo\varnothing}$. During training, $c_t$ is randomly replaced by $c_{mo\varnothing}$ with a predefined probability (i.e., $c_{mo\varnothing}$ is only updated when used). This enables joint learning of conditional and unconditional generation, enhancing robustness and controllability.

### 3.3   MODEL SCALING

Following the scaling law of generative models (Wang et al., 2025; Zhou et al., 2025; Cheng et al., 2025), we scale MTVCraft from 6B to 18B parameters. The 6B version uses CogVideoX-5B (Yang et al., 2024b) as the backbone, while the 18B version adopts Wan-2-1-14B (Wan et al., 2025), which provides significantly stronger visual generation capacity and is therefore better suited for animating diverse open-world characters. As shown in Figure 6, the scaled version introduces an additional text control branch compared with the 6B version in Figure 4, thus enabling joint control by both text and motion. We reuse 4DMoT and unconditional motion tokens without modification, which substantially reduces training cost. For 4D motion attention, since the motion tokens derived from codebook are of dimension 3072 while the vision tokens in Wan-2-1 have dimension 5120, we apply zero-padding along the channel dimension for alignment. This preserves the spatial-temporal structure of motion tokens faithfully without requiring additional transformations or retraining. Overall, MTVCraft demonstrates strong scalability and seamless integration of 4D motion tokens, indicating that our method is readily applicable to and versatile across different diffusion backbones in practice. More architectural details of MTVCraft-18B are provided in Appendix C.

## 4   EXPERIMENT

**Benchmarks**   Following (Zhou et al., 2025), we conduct experiments on TikTok (Jafarian & Park, 2021) and Fashion (Zablotskaia et al., 2019) benchmarks. The evaluation metrics are detailed in Appendix J. And the evaluation details are provided in Appendix A.3.

**Implementation Details**   For 4DMoT, we use a codebook of size 8192 with a code dimension of 3072. Quantization is performed with an exponential moving average update strategy using a decay constant of $\lambda = 0.99$. To maintain codebook utilization, unused codes are reset every 20 steps. The sliding window size is set to 8, and the commitment loss weight in Equation 2 is set to 0.25. For MV-DiT, we insert a 4D motion attention module every two DiT blocks. The condition drop probabilities are set to 0.2 for motion and 0.1 for text, respectively. The classifier-free guidance scale is set to 3.0 for motion conditions and 6.0 for text conditions. All experiments are conducted on 8 NVIDIA H100 GPUs. Additional training and architectural details are provided in Appendix A.

Table 1: Quantitative Results on TikTok (Jafarian & Park, 2021) Benchmark.

| Model | PSNR↑ | SSIM↑ | LPIPS↓ | FID↓ | FVD↓ | FID-VID↓ |
|---|---|---|---|---|---|---|
| MusePose (Hu, 2024) | 18.20 | 0.757 | 0.248 | 41.99 | 532.75 | 14.60 |
| MooraAA (Hu, 2024) | 18.62 | 0.764 | 0.230 | 37.28 | 501.22 | 12.40 |
| ControlNeXt (Peng et al., 2024) | 16.31 | 0.728 | 0.296 | 33.48 | 548.01 | 28.23 |
| Animate-X (Tan et al., 2024) | 16.71 | 0.743 | 0.285 | 32.77 | 508.63 | 17.47 |
| MimicMotion (Zhang et al., 2024a) | 19.30 | 0.751 | 0.220 | 34.88 | 472.51 | 9.30 |
| RealisDance-DiT (Zhou et al., 2025) | 17.55 | 0.717 | 0.261 | 30.39 | 458.81 | - |
| Unianimate-DiT (Wang et al., 2025) | 19.35 | 0.765 | 0.235 | 28.47 | 402.14 | 9.12 |
| MTVCraft-6B | 19.35 | 0.760 | 0.219 | 23.58 | 317.21 | 8.56 |
| MTVCraft-18B | **19.84** | **0.779** | **0.217** | **20.70** | **276.65** | **7.31** |

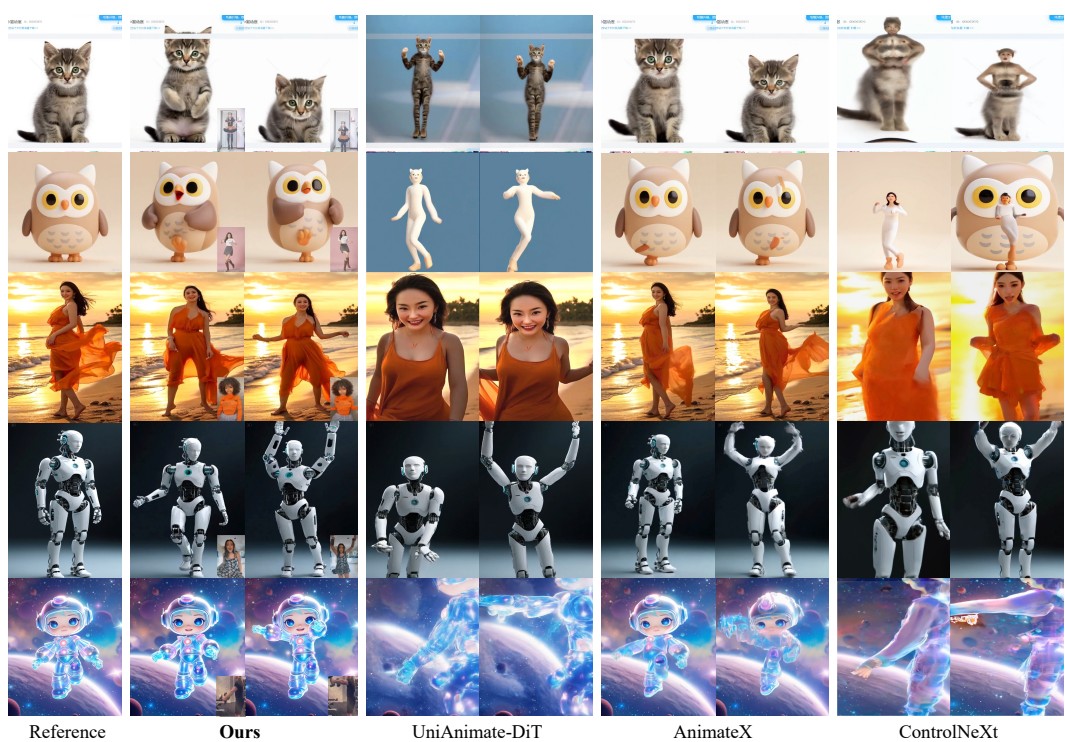

Reference     **Ours**     UniAnimate-DiT     AnimateX     ControlNeXt

Figure 5: **Qualitative Comparison.** Our MTVCraft consistently demonstrates the best motion transfer performance and high appearance consistency across various scenes and diverse characters.

## 4.1 SOTA COMPARISON

Figure 1, 5, 13, 15 demonstrate MTVCraft's superior animation performance in pose accuracy and identity consistency. MTVCraft exhibits strong zero-shot generalization in 4D worlds, handling full-body or half-body characters, across diverse styles and scenes. MTVCraft remains robust when the target pose is misaligned with the reference image (e.g., Owl in Figure 5), while other methods fail, highlighting effective disentanglement of motion from the driving video. Importantly, despite being trained exclusively on a human-centric dataset, MTVCraft is capable of animating non-human subjects, including animals and inanimate objects. This demonstrates the benefit of 4D motion representation via tokenization of differential joint coordinates. Table 1 shows state-of-the-art performance across all metrics, and additional experiments in Appendix G further confirm it. More qualitative comparisons and cases are provided in Appendix N and O.

Table 2: Ablation Study on TikTok (Jafarian & Park, 2021) Benchmark.

| Model | Choice | PSNR↑ | SSIM↑ | LPIPS↓ | FID↓ | FVD↓ | FID-VID↓ |
|-------|--------|-------|-------|--------|------|------|----------|
| 4D MT | w/o quantize | 18.76 | 0.732 | 0.226 | 24.04 | 332.97 | 9.39 |
|  | w/o differential motion | 19.08 | 0.740 | 0.223 | 24.37 | 325.40 | 8.92 |
|  | w/ 3D quantization | 19.12 | 0.742 | 0.221 | 23.94 | 329.86 | 9.04 |
| 4D MA | w/ dynamic PE | 17.23 | 0.733 | 0.247 | 28.24 | 383.22 | 11.85 |
|  | w/ learnable PE | 16.90 | 0.719 | 0.259 | 28.69 | 397.64 | 11.74 |
|  | w/ 1D temporal RoPE | 16.86 | 0.718 | 0.263 | 29.45 | 458.29 | 12.15 |
|  | w/ 2D spatial RoPE | 16.28 | 0.704 | 0.266 | 30.07 | 459.59 | 10.73 |
|  | w/ 3D spatial RoPE | 16.99 | 0.723 | 0.259 | 28.15 | 435.80 | 11.28 |
|  | w/o PE | 16.51 | 0.707 | 0.281 | 32.56 | 548.31 | 13.40 |
| | Our Default Design | **19.35** | **0.760** | **0.219** | **23.58** | **317.21** | **8.56** |

## 4.2 ABLATION STUDY

To validate our key designs, we conduct ablation studies on the 4D Motion Tokenizer and 4D Motion Attention. Table 2 shows the performance impact of modifying or removing specific components.

**4D Motion Tokenizer (MT)** (1) Without quantization, the VQ-VAE degenerates into a standard autoencoder producing continuous motion tokens, leading to degraded performance. This confirms that discrete motion tokens in a unified space help stabilize motion learning and improve generalization across diverse motion patterns. (2) Removing differential motion also worsens performance, as it prevents the model from explicitly modeling relative joint displacements, which are crucial for capturing fine-grained and temporally coherent motion dynamics across frames. (3) Compared with 3D quantization (t, x, y), adding the z-axis captures depth-aware geometric information and further improves overall results. We provide a more systematic analysis of our 4DMoT in Appendix E.

**4D Motion Attention (MA)** We explore various positional encoding (PE) designs for the motion attention module: (1) Dynamic PE computes RoPE using joint coordinates of the first frame, but performs poorly due to instability and training difficulties; (2) Learnable PE struggles to converge and fails to provide reliable positional cues; (3) 1D temporal RoPE, (4) 2D spatial RoPE (x, y), (5) 3D spatial RoPE (x, y, z) all fail to capture full 4D spatiotemporal dependencies, resulting in noticeable visual artifacts such as identity drift or temporal jittering; (6) w/o PE completely removes positional encoding and yields the worst performance (e.g., FVD 548.31 versus 317.21), highlighting the necessity of explicit positional information. Overall, these results demonstrate the importance of modeling 4D positional information. Additional ablation studies are provided in Appendix H.

## 5 CONCLUSION

We present MTVCraft, the first framework that directly tokenizes raw motion sequences for controllable character video generation. By integrating a 4D motion VQVAE with novel motion attention and 4D RoPE, MTVCraft enables precise 4D controllability and achieves state-of-the-art generalization to arbitrary characters, including non-human and AI-generated ones. Finally, its scalability to larger model sizes further enhances spatial-temporal coherence, identity fidelity, and controllable animation quality, paving the way for more versatile and realistic character image animation.

## ACKNOWLEDGMENTS

This work was supported by Guangdong Science and Technology Program (Grant No. 2024TQ08X365).

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

# Appendix

## TABLE OF CONTENTS

# A  MORE IMPLEMENTATION DETAILS

## A.1  MODEL ARCHITECTURAL DETAILS

**4DMoT**  Our 4D motion tokenizer consists of an encoder, a quantizer, and a decoder. Encoder begins with a convolutional input layer that projects the input channels from 3 to 32, followed by three ResNet blocks and downsampling blocks, with channel dimensions of (32, 128, 512), frame downsampling rates (2, 2, 1), and joint downsampling rates (1, 1, 1). A final convolutional output layer maps the features to a code dimension of 3072. Quantizer uses a codebook of size 8,192 with a code dimension of 3072. Decoder starts with a convolutional input layer that projects the code dimension from 3072 to 512, followed by three ResNet blocks and upsampling blocks symmetric to those in the encoder. A final convolutional output layer maps the features back to 3 channels. The overall number of trainable parameters in our 4DMoT is 48M.

**MV-DiT**  The 6B version is based on CogVideoX-5B-T2V (Yang et al., 2024b). To better support motion-centric generation, we remove the original text-processing branch and insert our proposed 4D Motion Attention layer after the self-attention layer in every two CogVideoX blocks. We train all the DiT blocks since the input channels are revised. The total number of trainable parameters in this small-scale model is approximately 6B, with all attention parameters randomly initialized. The 18B version is based on Wan-2-1-I2V-14B (Wan et al., 2025). In this version, the text-processing branch is retained, and the 4D Motion Attention layer is inserted after the cross-attention layer in every two Wan-2-1 blocks. We only train the newly added attention modules to better preserver the powerful generalization capability of the base model. The total number of trainable parameters in this large-scale model is approximately 4B, with all attention parameters randomly initialized.

## A.2  TRAINING DETAILS

**4DMoT**  The entire VQVAE model is trained from scratch using the AdamW optimizer with $\beta_1 = 0.9$, $\beta_2 = 0.99$, a weight decay of $1 \times 10^{-4}$, and a batch size of 32 per GPU (256 in total). We train for 200K iterations with a learning rate of $1 \times 10^{-4}$, followed by an additional 100K iterations with a reduced learning rate of $1 \times 10^{-5}$. During training, joint coordinates are randomly scaled and shifted within a range of 0–10% to augment diversity. The sampling frames of video and motion sequence are randomly chosen from $\{33, 49, 81, 97, 129\}$, with the stride randomly selected from $\{1, 2\}$. We adopt float32 precision to ensure stable codebook learning.

**MV-DiT**  We optimize both the 6B and 18B models using the AdamW optimizer with $\beta_1 = 0.9$, $\beta_2 = 0.99$, a weight decay of $1 \times 10^{-2}$, and a batch size of 4 per GPU (32 in total). The models are trained for 30K iterations (approximately 12 H100 days) with a learning rate of $2 \times 10^{-5}$ and 300 warm-up steps. During training, videos are resized and randomly cropped to the nearest resolution bucket, exposing the model to a range of resolutions and aspect ratios to improve generalization across diverse video qualities and sizes. The video sampling frames are randomly chosen from $\{33, 49, 81, 97, 129\}$, with the stride randomly selected from $\{1, 2\}$. We adopt bfloat16 precision for efficient training and DeepSpeed ZeRO-2 (Rasley et al., 2020) to reduce memory consumption.

## A.3  EVALUATION DETAILS

We follow the evaluation settings of Realisdance-DiT (Zhou et al., 2025). We adopt DDIM (Song et al., 2020) for sampling, performing 50 inference steps. The motion classifier-free guidance scale is set to 3.0. The text prompt is always set to "a person is dancing" and the text classifier-free guidance scale is set to 6.0. Since some methods, e.g., (Tan et al., 2024; Wang et al., 2025), do not report the FID-VID (Balaji et al., 2019) metric in their original papers, we re-evaluate them under the same settings. For other reported metrics, we directly use the values from the respective papers.

# B  MORE DETAILS OF DATASET CURATION

**Shot Segmentation**  We use AutoShot (Zhu et al., 2023), an automated shot boundary detection algorithm, to detect shot boundaries and segment raw videos into coherent, temporally continuous

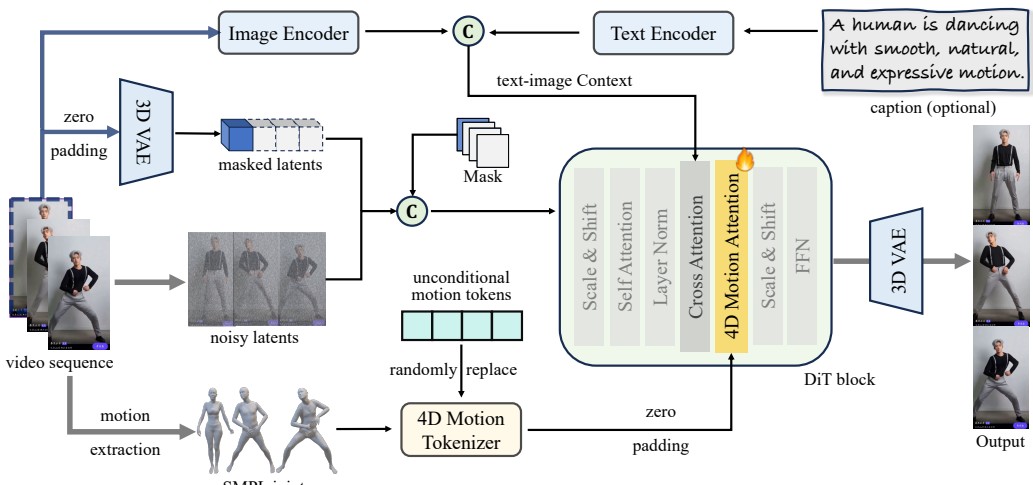

Figure 6: **Architecture of MTVCraft-18B.** To demonstrate the versatility of our approach and further improve performance, we scale the model to a larger DiT and enable joint text-motion control. Here, zero-padding aligns the motion token dimension with the DiT hidden dimension.

shots. This is critical to eliminate abrupt scene changes and ensure that each clip maintains smooth temporal coherence, providing a reliable foundation for subsequent quality filtering operations.

**Pose Estimation** For each segmented clip, we use NLF-Pose (Sárándi & Pons-Moll, 2024) to estimate frame-wise SMPL (Loper et al., 2023) joint parameters, along with the corresponding confidence scores for each joint. These confidence scores reflect the uncertainties of the predicted poses, providing a measure of reliability for downstream processing. During pose estimation, the camera is fixed with a 55° field of view along the image's longer side and a centered principal point.

**Single-person Sub-clip Extraction** Unfortunately, the current version of MTVCraft cannot support multi-person animations with different poses. Thus, we focus on extracting continuous sub-clips (more than 33 frames) from each video containing only a single human pose with valid predictions across all frames. In other words, frames with no pose or multiple poses detected are excluded.

**Pose Uncertainty Filtering** We compute the average of the maximum joint uncertainty across all frames for each video and discard the top 10% most uncertain videos, as they typically contain extreme pose errors. Additionally, we manually inspected 200 randomly sampled clips from the remaining subset and observed highly accurate motion estimations. When projected back to 2D, the 3D poses show strong visual alignment with the 2D keypoints detected by DW-Pose (Yang et al., 2023). While a few poses may still exhibit minor imperfections (e.g., occasional missing frames), such diversity, as a data augmentation technique, enhances the tokenizer's robustness.

**Visual and Motion Quality Assessment** For the remaining clips, we evaluate four complementary metrics to assess overall quality: (1) Aesthetic score: we use the LAION-Aesthetics predictor (Schuhmann et al., 2022), which is a linear estimator built on top of CLIP (Radford et al., 2021), to predict the aesthetic quality of images. (2) Optical flow magnitude: we use the UniMatch model (Xu et al., 2023) to compute the optical flow between frames, assessing the extent of motion. (3) Laplacian blur score: we apply the Laplacian operator using OpenCV [1] to detect blurry frames. (4) OCR text ratio: we use CRAFT (Baek et al., 2019) to detect text regions and estimate the proportion of text within each frame, filtering out clips dominated by textual content.

The thresholds for these metrics are set to 5.0, 2.0, 100, and 0.05, respectively. Clips that fail to meet any of the above quality thresholds are discarded. Through this rigorous filtering process, we

---

[1]OpenCV: `https://docs.opencv.org/3.4/d5/db5/tutorial_laplace_operator.html`

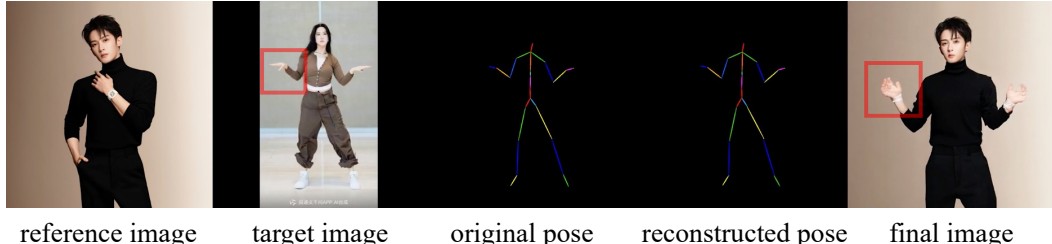

| reference image | target image | original pose | reconstructed pose | final image |

Figure 7: **Failure Case.** Precise hand control is challenging due to the lack of hand supervision.

obtain a final dataset of 30K high-quality motion video clips featuring clear frames with minimal textual content, continuous and consistent single-person motions, and smooth temporal transitions. Finally, we manually inspected a subset of clips to confirm the overall quality of the curated dataset.

## C  More Details of Model Scaling

As shown in Figure 6, the scaled 18B model introduces an additional text-image joint control branch, which concatenates the reference image features from the first frame with text embeddings. These fused features are then used to perform cross-attention with the vision latents, allowing the model to jointly reason over textual instructions and visual appearance. This mechanism enables fine-grained text control over attributes such as identity, motion, and style, enhancing semantic-level control over generation and improving consistency with the reference frame across time.

To preserve identity from the reference image, we adopt the same strategy as Wan-2-1-I2V. Specifically, we pad the reference frame $x_0$ temporally with all-zero frames to match the length of the target video. The sequence is then encoded into latent representations $z_{\text{ref}} \in \mathbb{R}^{f \times c \times h \times w}$ using a frozen 3D VAE. These latents are concatenated with a binary mask and the noisy video latents $\{z_t\}_{t=1}^{f}$ to construct the composite vision latents $z_{\text{vision}}$, which are defined as:

$$z_{\text{ref}} = \text{VAE}(\text{ZeroPad}(x_0)) \tag{9}$$

$$z_{\text{vision}} = \text{Concat}(z_{\text{ref}}, \text{mask}, z_t) \in \mathbb{R}^{f \times 2c \times h \times w} \tag{10}$$

Here, the binary mask (with 4 channels) accounts for the VAE's temporal compression rate of 4. The resulting $z_{\text{vision}}$ is then patchified and projected into the attention space, enabling interaction with the noisy latents and facilitating identity preservation during self-attention in the DiT blocks.

## D  Limitation and Discussion

While MTVCraft achieves impressive performance across diverse scenarios, it still presents certain limitations. First, the model may generate inaccurate results when the base model cannot handle well. Second, as shown in Figure 7, precise hand articulation remains a challenge, as clear and detailed hand motion is underrepresented in our SMPL motion-video dataset.

In addition to the technical limitation, we recognize broader concerns in the use of MTVCraft, such as potential misuse involving unauthorized identity manipulation or violation of data copyrights, especially when animating reference images sourced from social platforms. MTVCraft must not be misused to fabricate harmful, misleading, or disrespectful content, such as mocking individuals or distorting cultural heritage. We request the responsible use of MTVCraft and plan to adopt safeguards such as user consent verification and watermarking, especially in public-facing applications.

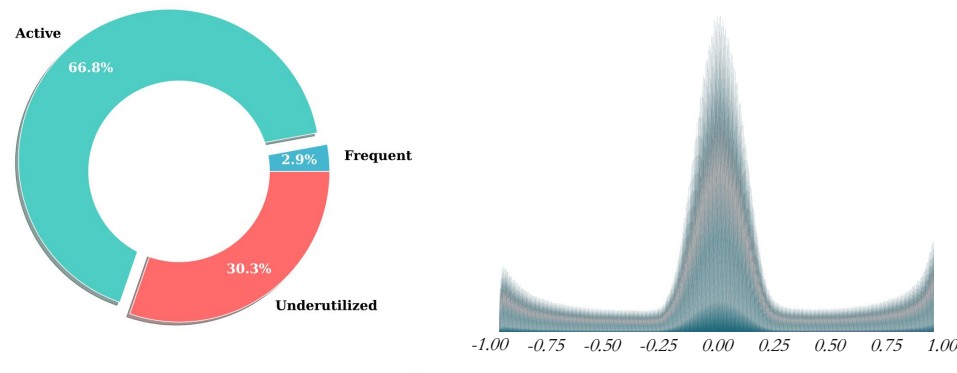

(a) Distribution of code usage rates       (b) Distribution of pairwise code similarity

Figure 8: **Quantitative Analysis of Codebook.** The left panel demonstrates that nearly 70% of the codes remain active during inference, indicating efficient utilization of the encoding space. The right figure shows that the cosine similarity of most code pairs is close to 0, confirming the model's ability to learn a discrete latent space characterized by highly decorrelated representations.

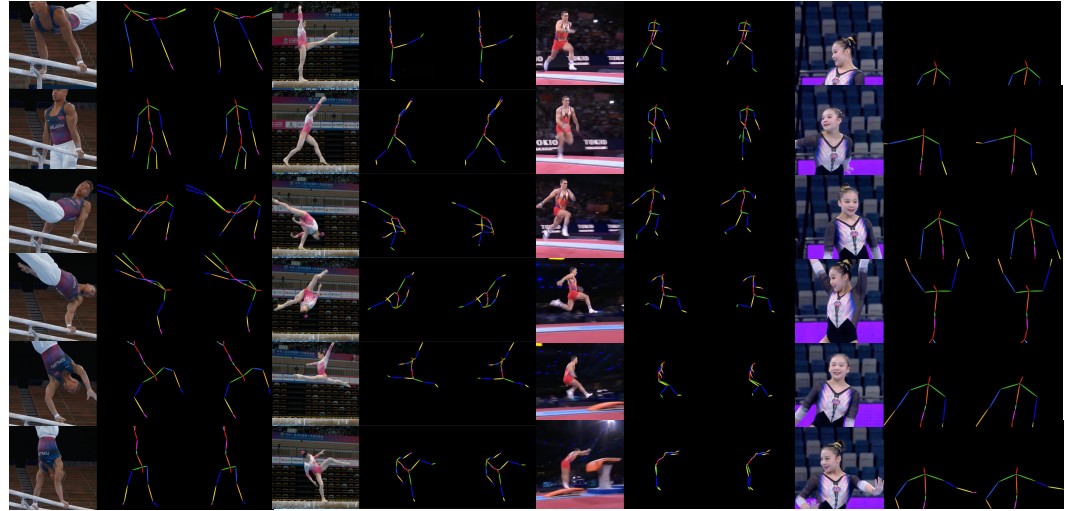

Figure 9: **Reconstruction Performance of 4DMoT on Unseen Gymnastics Data.** Each group consists of three images: the original image (first column), the extracted original pose (second column), and the reconstructed pose (third column). All poses are visualized as 3D joint skeletons, projected into 2D image space using joint coordinates. Our motion VQVAE demonstrates strong generalization to unseen motion data and achieves accurate and robust reconstruction quality.

# E  MORE DETAILS OF 4D MOTION TOKENIZER

## E.1  TOKENIZING STRATEGY: DISCRETE VS. CONTINUOUS

We adopt discrete tokens instead of continuous ones for four main reasons. (1) Quantization suppresses low-level noise and encourages the model to capture high-level semantic motion patterns. This is similar to how tokenization benefits language models. (2) A unified codebook allows motion sequences to be represented by a compact set of motion primitives, which improves generalization to unseen motions while avoiding redundant variability. (3) Discrete tokens can be seamlessly integrated into DiT through embedding lookups, which makes training more efficient and stable compared to handling high-dimensional continuous trajectories. (4) Prior motion generation works like (Zhang et al., 2025; Hosseyni et al., 2025) also leverage discrete tokens, but they are defined in

SMPL-parameter space. By instead tokenizing joint coordinates, our approach provides more direct positional information aligned with pixel-level generation, thereby facilitating effective learning.

## E.2 TOKENIZATION VS. RENDERING IMAGES

We adopt tokenization rather than rendering images as motion conditions for two key reasons. First, tokenization enables the model to directly learn semantic 4D motion information. For example, in 2D renderings it is difficult to distinguish whether a person appears small due to actual body size or because they are far from the camera. In contrast, tokenized joint coordinates explicitly preserve the depth dimension, providing unambiguous and compact motion cues. Second, tokenization decouples motion from absolute position and shape variations, thereby reducing the risk of overfitting to specific human appearances. Rendered images, by contrast, inherently encode biases such as position, scale, and limb length, which may lead to poor generalization across diverse characters and viewpoints. Overall, tokenization provides a robust, interpretable, and geometry-aware representation of motion, which is better suited for controllable video generation than image renderings.

## E.3 ANALYSIS OF CODEBOOK

To evaluate the efficiency of our codebook utilization, we conduct statistical inference on a test set comprising randomly selected 6400 motion samples. The codes are categorized into three usage frequency levels based on predefined thresholds: underutilized ($<1\%$), active ($1\%$-$15\%$), and frequent ($>15\%$) as shown in Figure 8 (a). The low frequency of frequent codes (2.9%) reflects an efficient selection of core features for reconstruction, minimizing overfitting to local training patterns. The broad distribution of active codes (66.8%) ensures expressive diversity, allowing the model to capture a wide range of patterns and preventing homogenization of the reconstruction. Meanwhile, moderate redundancy of underutilized codes (30.3%) improves the robustness of the tokenziation process, allowing the model to support richer feature combinations. This percent (i.e., 30.0%) is much lower than the extreme unused code rates in VQ-VAE (e.g. 50% with large codebook size (Zhu et al., 2024a)). This balanced utilization pattern validates our effective codebook optimization, showing that it achieves both compactness and diversity while avoiding codebook collapse.

Moreover, we conduct a comprehensive analysis of the codebook's latent space to assess the diversity of its entries. Specifically, we computed pairwise cosine similarities between codes and visualized their distribution as shown in Figure 8 (b). The results show that most code pairs exhibit near-zero similarity, indicating significantly uncorrelated characteristics. This finding confirms that the model successfully constructs a discrete latent space with high representational independence.

## E.4 RECONSTRUCTION PERFORMANCE

Finally, to directly assess the effectiveness, we evaluate the reconstruction quality of the motion VQVAE on unseen gymnastics motion sequences, which represent a challenging and highly dynamic test case. As illustrated in Figure 9, our model can accurately reconstruct complex human poses, even in highly dynamic motion scenarios. All results are visualized as 3D joint skeletons rendered in 2D image-pixel space. The reconstructed poses closely match the original inputs, demonstrating the VQVAE's strong generalization capability and its ability to preserve spatial-temporal structure.

The results from these three experimental groups collectively demonstrate the effectiveness and suitability of the proposed 4DMoT for the downstream character image animation task.

## F MORE DETAILS OF 4D MOTION RoPE

Positional encoding is critical for modeling spatial-temporal dependencies. Removing it leads to significant performance degradation, consistent with prior observations in 3D RoPE for CogVideoX and Wan. In our study, we explore several positional encoding strategies and find that 4D RoPE is best suited for 4D motion tokens. Without it, the model struggles to converge. In this section, we provide detailed implementation and analysis of our proposed 4D RoPE design.

---

**Algorithm 1** 4D RoPE of Motion Tokens

---

**Require:** Dataset-wide mean joint positions `mean_joints` $\in \mathbb{R}^{J \times 3}$, number of latent frames $T$ after 4× downsampling, and attention head dimension $D$.

1. Extract spatial coordinates:
   $x \leftarrow$ `mean_joints`$[:, 0]$
   $y \leftarrow$ `mean_joints`$[:, 1]$
   $z \leftarrow$ `mean_joints`$[:, 2]$
   $t \leftarrow \{0, 1, \ldots, T-1\}$
2. Centralize spatial positions:
   $\hat{x} \leftarrow x - \mathrm{mean}(x)$
   $\hat{y} \leftarrow y - \mathrm{mean}(y)$
   $\hat{z} \leftarrow z - \mathrm{mean}(z)$
3. Compute 1D RoPE for each axis (see Equation 1):
   $(\cos_t, \sin_t) \in R^{T \times (D/4) \times 2} \leftarrow \texttt{RoPE}(t, D/4)$
   $(\cos_x, \sin_x) \in R^{J \times (D/4) \times 2} \leftarrow \texttt{RoPE}(\hat{x}, D/4)$
   $(\cos_y, \sin_y) \in R^{J \times (D/4) \times 2} \leftarrow \texttt{RoPE}(\hat{y}, D/4)$
   $(\cos_z, \sin_z) \in R^{J \times (D/4) \times 2} \leftarrow \texttt{RoPE}(\hat{z}, D/4)$
4. Broadcast time RoPE over all joints:
   $(\cos_t, \sin_t) \in R^{T \times J \times (D/4) \times 2} \leftarrow \texttt{Repeat}((\cos_t, \sin_t), \mathrm{dim} = 1, \mathrm{repeats} = J)$
5. Broadcast joint RoPE over all frames:
   $(\cos_x, \sin_x) \in R^{T \times J \times (D/4) \times 2} \leftarrow \texttt{Repeat}((\cos_x, \sin_x), \mathrm{dim} = 0, \mathrm{repeats} = T)$
   $(\cos_y, \sin_y) \in R^{T \times J \times (D/4) \times 2} \leftarrow \texttt{Repeat}((\cos_y, \sin_y), \mathrm{dim} = 0, \mathrm{repeats} = T)$
   $(\cos_z, \sin_z) \in R^{T \times J \times (D/4) \times 2} \leftarrow \texttt{Repeat}((\cos_z, \sin_z), \mathrm{dim} = 0, \mathrm{repeats} = T)$
6. Concatenate positional encodings across channel dimensions:
   `freqs_cos` $\leftarrow \texttt{Concat}(\cos_t, \cos_x, \cos_y, \cos_z)$
   `freqs_sin` $\leftarrow \texttt{Concat}(\sin_t, \sin_x, \sin_y, \sin_z)$
   **return** `freqs_cos, freqs_sin`

---

## F.1  4D RoPE Design and Implementation

To enhance the spatial-temporal relationships of 4D motion tokens, we design the 4D RoPE. For each motion token, we compute its positional encoding based on the corresponding 4D coordinates $(t, x, y, z)$, where $t$ denotes the frame index. The spatial coordinates $(x, y, z)$ are centralized by subtracting the global mean joint position, which is computed over the entire dataset by averaging all joints across all frames along the joint axis. This centralization ensures that the positional encoding remains consistent and invariant to global spatial shifts. For each of the four dimensions $(t, x, y, z)$, we compute sinusoidal RoPE features independently according to Equation 1, with each contributing a quarter of the total attention head dimension, i.e., $D/4$. Temporal RoPE features are then broadcast across all joints, while spatial RoPE features are broadcast across all frames. This ensures that each motion token is equipped with the corresponding and structured 4D positional encoding, enabling precise modeling of both motion dynamics and spatial structure. The detailed procedure of 4D motion RoPE is provided in Algorithm 1. For vision tokens, we use the image height and width as the $x$ and $y$ coordinates, respectively, and set $z$ to zero (representing no depth variation). This design enables natural interaction between vision and motion tokens in 4D space.

## F.2  Visualization and Analysis of 4D RoPE

To visualize the advantages of our proposed design, Figure 10 (a) presents the cross-attention maps in different attention layers. The vertical axis represents motion tokens across frames and joints, while the horizontal axis corresponds to vision tokens across frames and pixels. To better visualize the impact, we resize the images to a resolution of $512 \times 512$. When positional encoding is omitted, attention maps tend to be structureless, indicating difficulty in capturing useful relationships. In contrast, when our 4D Rotary Position Embedding (RoPE) is applied, the attention patterns become increasingly structured across layers, suggesting that the model benefits from explicit spatial-temporal positional cues, enabling effective interaction between vision and motion representations.

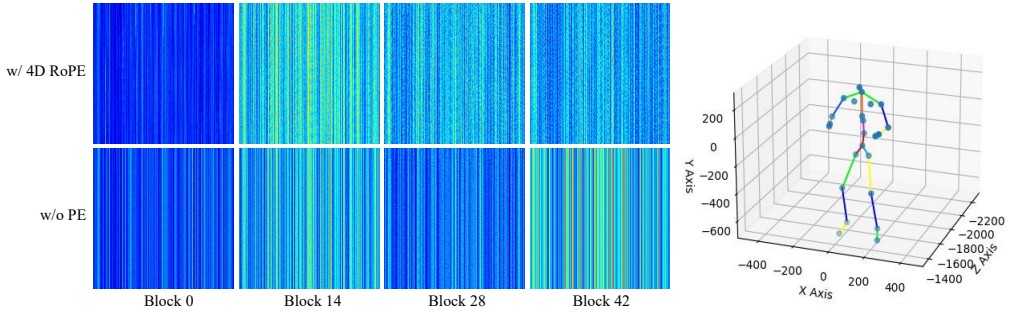

(a) Visualization of cross-attention maps with and without 4D RoPE     (b) Visualization of dataset mean used in 4D RoPE

Figure 10: **Effectiveness of 4D RoPE for Motion-Vision Interaction.** (a) Cross-attention maps at different Transformer blocks show that 4D RoPE enables structured interactions between motion and vision tokens. (b) Visualization of mean joint coordinates across the dataset, used to compute 4D RoPE, providing typical spatial cues that facilitate consistent cross-modal modulation.

Furthermore, we visualize the mean joint positions of the dataset used in our 4D RoPE design. As shown in Figure 10 (b), the result exhibits a standard human skeleton composed of 3D joint coordinates. These averaged joint positions serve as the spatial information for the 4D RoPE calculation, enabling the model to encode relative spatial relationships effectively and consistently across different motion sequences. This design not only enhances the robustness of cross-modal interaction but also aligns motion and vision tokens in a physically plausible manner.

# G   MORE QUANTITATIVE EXPERIMENTS

Table 3 presents the quantitative results on Fashion (Zablotskaia et al., 2019) test set. Our MTVCraft consistently achieves superior scores across all metrics, demonstrating its effectiveness in modeling motion and preserving identity. These results further validate the advantages of directly leveraging 4D motion tokenization over conventional pose-rendered image based methods.

# H   MORE ABLATION STUDY

**Ablation on Fashion Dataset**   Table 4 summarizes the ablation results. Consistent with the conclusions derived from Table 2, the default design achieves the best performance. This highlights the crucial role of both the discrete differential motion representation and explicit 4D positional encoding in stabilizing motion learning and improving generalization.

**Ablation of Motion-aware CFG**   Figure 11 presents the qualitative and quantitative evaluations of our motion-aware CFG scale. On TikTok (Jafarian & Park, 2021) benchmark, a CFG scale of 3.0 yields the best performance, particularly for the FVD metric. For the FID-VID metric, the scale appears to have minimal impact. For visual comparisons on the right, increasing the CFG scale enhances pose alignment, but it also introduces more artifacts and potentially degrades quality.

**Additional Ablation**   For MTVCraft-18B, we further examined several alternative designs. Replacing zero-padding with a linear or MLP layer did not converge within 10K steps, suggesting that the simple projection is insufficient to stabilize large-scale motion learning. We also attempted to inject a pretrained SMPL-parameter space tokenizer (Guo et al., 2024) into the DiT backbone, but training collapsed in the early stages. These observations justify the necessity of our discrete differential motion representation (i.e., joint coordinates) and explicit 4D positional encoding.

# I   TRAINING CURVES

As shown in Figure 12, we plot the training loss curves of our 4DMoT and MV-DiT. 4DMoT exhibits rapid convergence, with the loss quickly decreasing early in training. In contrast, MV-DiT

Table 3: Quantitative Results on Fashion (Zablotskaia et al., 2019) Benchmark.

| Model | PSNR↑ | SSIM↑ | LPIPS↓ | FID↓ | FVD↓ | FID-VID↓ |
|---|---|---|---|---|---|---|
| MusePose (Hu, 2024) | 22.20 | 0.896 | 0.067 | 14.95 | 96.17 | 10.94 |
| MooraAA (Hu, 2024) | 20.83 | 0.880 | 0.093 | 27.74 | 149.66 | 10.13 |
| ControlNeXt (Peng et al., 2024) | 18.48 | 0.853 | 0.132 | 13.82 | 143.02 | 14.60 |
| Animate-X (Tan et al., 2024) | 22.15 | 0.893 | 0.069 | 10.11 | 70.47 | 11.87 |
| MimicMotion (Zhang et al., 2024a) | 23.80 | 0.913 | 0.061 | 15.40 | 80.89 | 8.17 |
| RealisDance-DiT (Zhou et al., 2025) | 23.33 | 0.908 | **0.053** | 10.81 | 72.94 | - |
| Unianimate-DiT (Wang et al., 2025) | 23.52 | 0.907 | 0.060 | 12.79 | 88.36 | 6.12 |
| MTVCraft-6B | 23.42 | 0.917 | 0.059 | 8.91 | 67.42 | 4.56 |
| MTVCraft-18B | **23.90** | **0.923** | 0.057 | **8.74** | **64.88** | **4.41** |

Table 4: Ablation Study on Fahsion (Zablotskaia et al., 2019) Benchmark.

| Model | Choice | PSNR↑ | SSIM↑ | LPIPS↓ | FID↓ | FVD↓ | FID-VID↓ |
|---|---|---|---|---|---|---|---|
| 4D MT | w/o quantize | 22.95 | 0.897 | 0.066 | 9.65 | 70.23 | 5.09 |
| | w/o differential motion | 23.18 | 0.905 | 0.067 | 9.12 | 68.51 | 4.87 |
| | w/ 3D quantization | 22.84 | 0.884 | 0.070 | 10.23 | 69.60 | 5.24 |
| 4D MA | w/ dynamic PE | 21.17 | 0.839 | 0.081 | 11.40 | 78.25 | 6.97 |
| | w/ learnable PE | 20.84 | 0.821 | 0.096 | 12.38 | 89.22 | 7.45 |
| | w/ 1D temporal RoPE | 20.56 | 0.801 | 0.105 | 14.07 | 98.61 | 8.17 |
| | w/ 2D spatial RoPE | 20.38 | 0.813 | 0.099 | 14.25 | 96.38 | 8.55 |
| | w/ 3D spatial RoPE | 20.77 | 0.812 | 0.108 | 13.95 | 99.48 | 8.34 |
| | w/o PE | 19.29 | 0.763 | 0.124 | 16.65 | 113.24 | 10.26 |
| | Our Default Design | **23.42** | **0.917** | **0.059** | **8.91** | **67.42** | **4.56** |

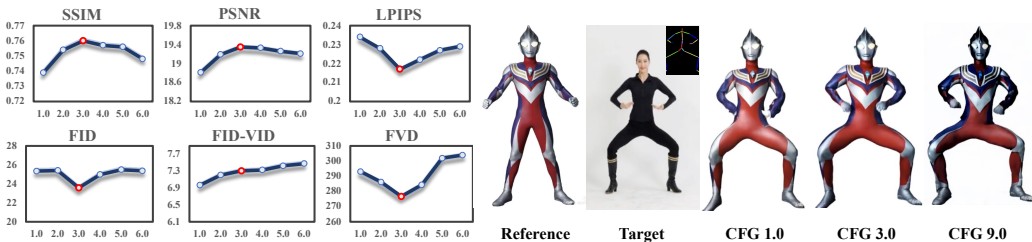

Figure 11: **Ablation of Motion-aware CFG.** A higher CFG scale leads to better pose alignment, but also introduces more artifacts. In our experiments, a scale of 3.0 achieves the best trade-off.

shows oscillatory convergence, with fluctuations in the loss curve before stabilizing. This difference highlights the distinct training dynamics of the two models, with the light motion tokenizer achieving a faster convergence while the relatively heavy DiT model requires more refinement for stable learning. After training for 100 epochs, both models performed exceptionally well.

## J  EVALUATION METRICS

In this section, we provide detailed formulations of our evaluation metrics.

- **PSNR** (Hore & Ziou, 2010):
  First, we compute the mean squared error for each frame:

$$\text{MSE} = \frac{1}{HWC} \sum_{x=1}^{W} \sum_{y=1}^{H} \sum_{c=1}^{C} \big(I_t(x,y,c) - \hat{I}_t(x,y,c)\big)^2. \tag{11}$$

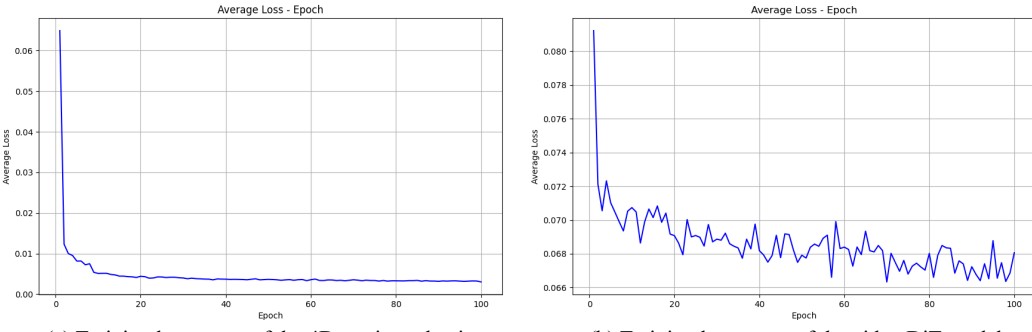

(a) Training loss curve of the 4D motion tokenizer.  (b) Training loss curve of the video DiT model.

Figure 12: Training loss curves of the 4D motion tokenizer and 4D motion-guided video DiT model. The tokenizer demonstrates smooth convergence with decreasing reconstruction and commitment loss, while the video DiT model gradually learns motion-aware video generation.

Then, we define Peak signal-to-noise ratio as:

$$\text{PSNR} = 20 \log_{10}\left( \frac{255}{\sqrt{\text{MSE}}} \right). \tag{12}$$

- **SSIM** (Wang et al., 2004):
  For each frame $t$ and color channel $c$, we first measure structural similarity:

$$\text{SSIM}_t^c = \text{SSIM}\big(I_t(\cdot, \cdot, c), \hat{I}_t(\cdot, \cdot, c)\big). \tag{13}$$

  Then, structural similarity index is the average score across all frames and channels:

$$\text{SSIM} = \frac{1}{TC} \sum_{t=1}^{T} \sum_{c=1}^{C} \text{SSIM}_t^c. \tag{14}$$

- **LPIPS** (Zhang et al., 2018):
  For each video frame, let $\phi_\ell(\cdot)$ denote the feature map from the $\ell$-th layer of a pretrained backbone network (e.g., AlexNet (Krizhevsky et al., 2012)), and let $w_\ell$ denote the learned channel-wise weights. The frame-level perceptual distance is defined as:

$$l_t = \frac{1}{L} \sum_{\ell=1}^{L} \frac{1}{H_\ell W_\ell} \big\| w_\ell \odot \big( \phi_\ell(I_t) - \phi_\ell(\hat{I}_t) \big) \big\|_1, \tag{15}$$

  where $H_\ell$ and $W_\ell$ denote the spatial dimensions of the $\ell$-th feature map. The video-level LPIPS score is then computed by averaging across all frames:

$$\text{LPIPS} = \frac{1}{T} \sum_{t=1}^{T} l_t. \tag{16}$$

- **FID** (Heusel et al., 2017):
  We measure frame-level visual quality using activations from a pretrained Inception-V3 network (Szegedy et al., 2016). Let $(\mu_r^{\text{fid}}, \Sigma_r^{\text{fid}})$ and $(\mu_f^{\text{fid}}, \Sigma_f^{\text{fid}})$ denote the Gaussian statistics of real and generated frames, respectively. The FID score is then defined as:

$$\text{FID} = \|\mu_r^{\text{fid}} - \mu_f^{\text{fid}}\|_2^2 + \text{Tr}\big(\Sigma_r^{\text{fid}} + \Sigma_f^{\text{fid}} - 2(\Sigma_r^{\text{fid}}\Sigma_f^{\text{fid}})^{1/2}\big). \tag{17}$$

- **FVD** (Unterthiner et al., 2018):
  We assess temporal realism using features extracted from the Inflated 3D ConvNet (I3D) (Carreira & Zisserman, 2017). Videos are split into non-overlapping 16-frame segments. Let $(\mu_r^{\text{fvd}}, \Sigma_r^{\text{fvd}})$ and $(\mu_f^{\text{fvd}}, \Sigma_f^{\text{fvd}})$ be the Gaussian statistics of real and generated video segments,

$$\text{FVD} = \|\mu_r^{\text{fvd}} - \mu_f^{\text{fvd}}\|_2^2 + \text{Tr}\big(\Sigma_r^{\text{fvd}} + \Sigma_f^{\text{fvd}} - 2(\Sigma_r^{\text{fvd}}\Sigma_f^{\text{fvd}})^{1/2}\big). \tag{18}$$

- **FVD-VID** (Balaji et al., 2019):
  To capture long-range temporal consistency, we aggregate I3D embeddings by averaging all clip-level features within each video, yielding a single descriptor per sequence. The statistics $(\mu_r, \Sigma_r)$ and $(\mu_f, \Sigma_f)$ are then estimated over these video-level descriptors. The distance is defined as:

$$\text{FVD-VID} = \|\mu_r - \mu_f\|_2^2 + \text{Tr}\big(\Sigma_r + \Sigma_f - 2(\Sigma_r\Sigma_f)^{1/2}\big). \tag{19}$$

## K    ETHICS STATEMENT

This work is conducted with the aim of advancing research in controllable video generation. We acknowledge that all experiments and analyses are performed in compliance with the ICLR Code of Ethics [2]. Besides, we certify that this submission complies with the submission instructions as described on `https://iclr.cc/Conferences/2026/AuthorGuide`. We claim that MTVCraft is intended solely for ethical research and creative applications, and acknowledge potential risks such as misuse for identity manipulation or misleading content.

## L    REPRODUCIBILITY STATEMENT

The overall model design is described in Section 3, while architectural details, hyperparameters, and optimization settings are provided in Section 4 and further elaborated in Appendix A. Additionally, we release all the codes and show many cases in Supplementary Material. These resources provide the necessary methodological details and ensures that readers can reliably reproduce our results.

## M    USE OF LARGE LANGUAGE MODELS

After completing the writing of this paper, we employed a large language model (LLM), e.g., ChatGPT-5 [3], solely for language-focused grammar and style proofreading. Specifically, we provided the PDF version to the LLM to identify and correct potential grammatical or typographical errors. All technical content, ideas, and experimental results were produced by the authors.

## N    MORE QUALITATIVE COMPARISONS

In this section, we provide additional qualitative comparisons to further demonstrate the effectiveness and robustness of our MTVCraft across a wide range of scenarios, character appearances, and motion types. As shown in Figure 13 and 15, our MTVCraft consistently demonstrates the best performance with high-quality character motion and high-fidelity appearance across different styles.

## O    MORE VISUALIZATION RESULTS

In this section, we provide additional visualizations. Figure 14 demonstrates our powerful zero-shot generalization to unseen diverse characters or even objects. Figure 16 provides more cases showing that we are able to transfer complex 4D motion sequences to unseen subjects. Figure 17 shows additional open-world character animations conditioned on different motion sequences, where MTVCraft consistently achieves high identity consis- tency and motion accuracy across various styles. Figure 18 shows human character animations conditioned on different motion sequences, where MTVCraft perfectly preserves both identity and motion accuracy across diverse real human characters. We also provide many videos generated by our MTVCraft in Supplementary Material.

---

[2]ICLR Code of Ethics: `https://iclr.cc/public/CodeOfEthics`
[3]Chat-GPT: `https://chatgpt.com`

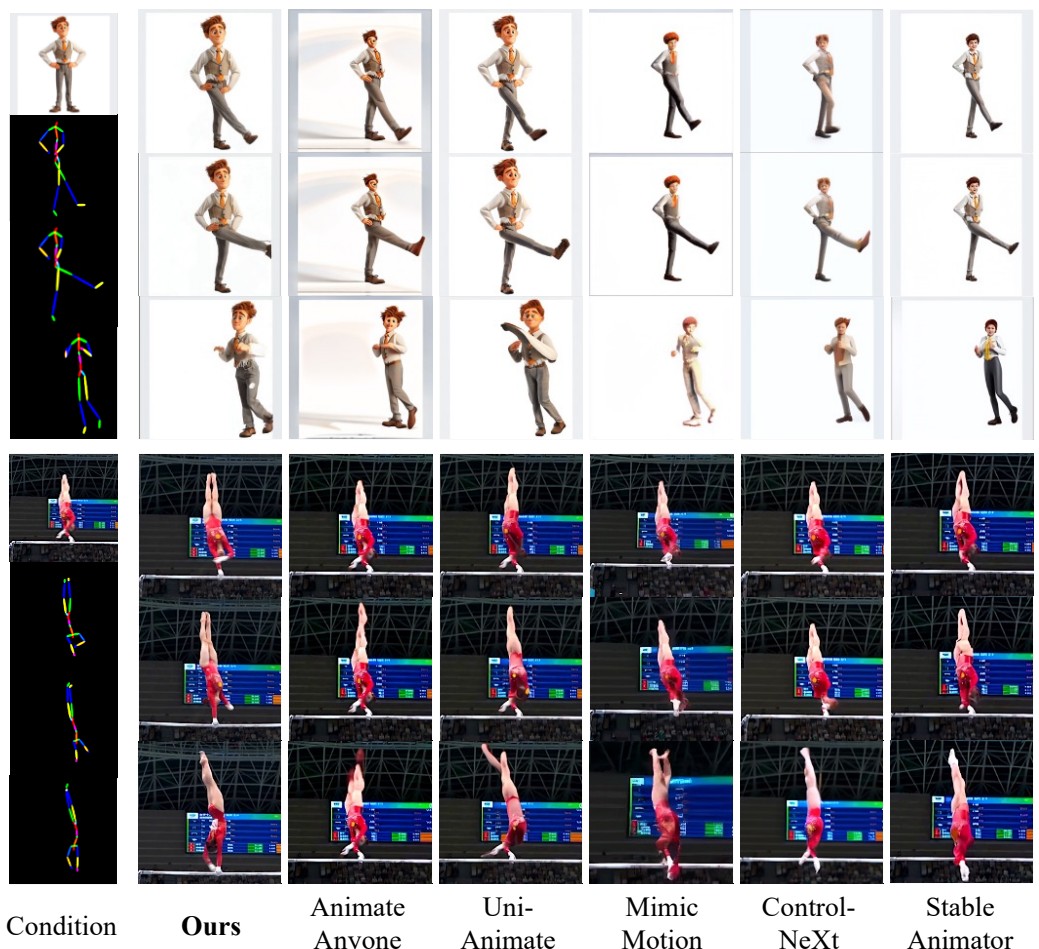

| Condition | **Ours** | Animate Anyone | Uni-Animate | Mimic Motion | Control-NeXt | Stable Animator |

Figure 13: **More Comparisons (1).** Our MTVCraft consistently demonstrates the best performance with high-quality human motion and high-fidelity appearance across different styles and scenes.

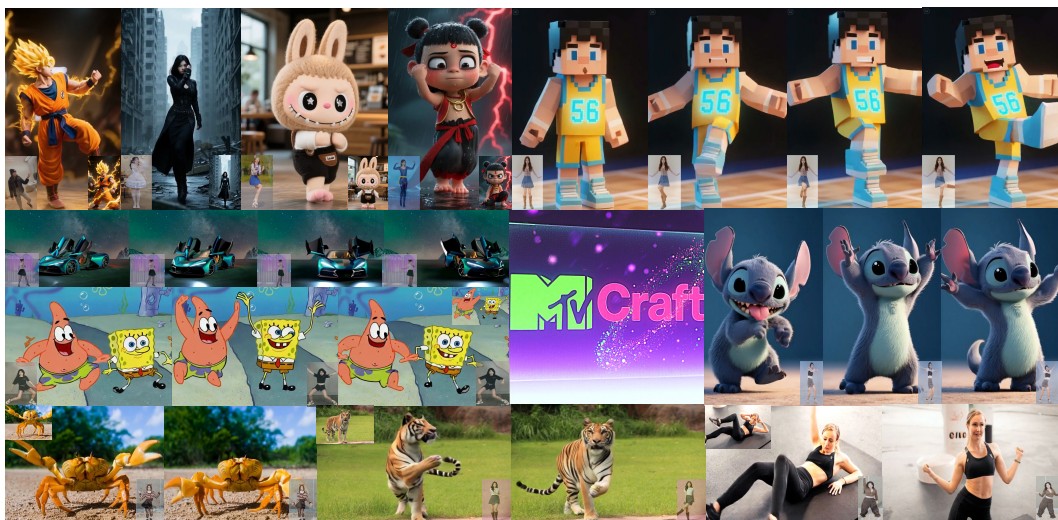

Figure 14: **More Visualization Results (1).** These cases demonstrate MTVCraft's powerful zero-shot generalization to unseen diverse characters or even animals, objects.

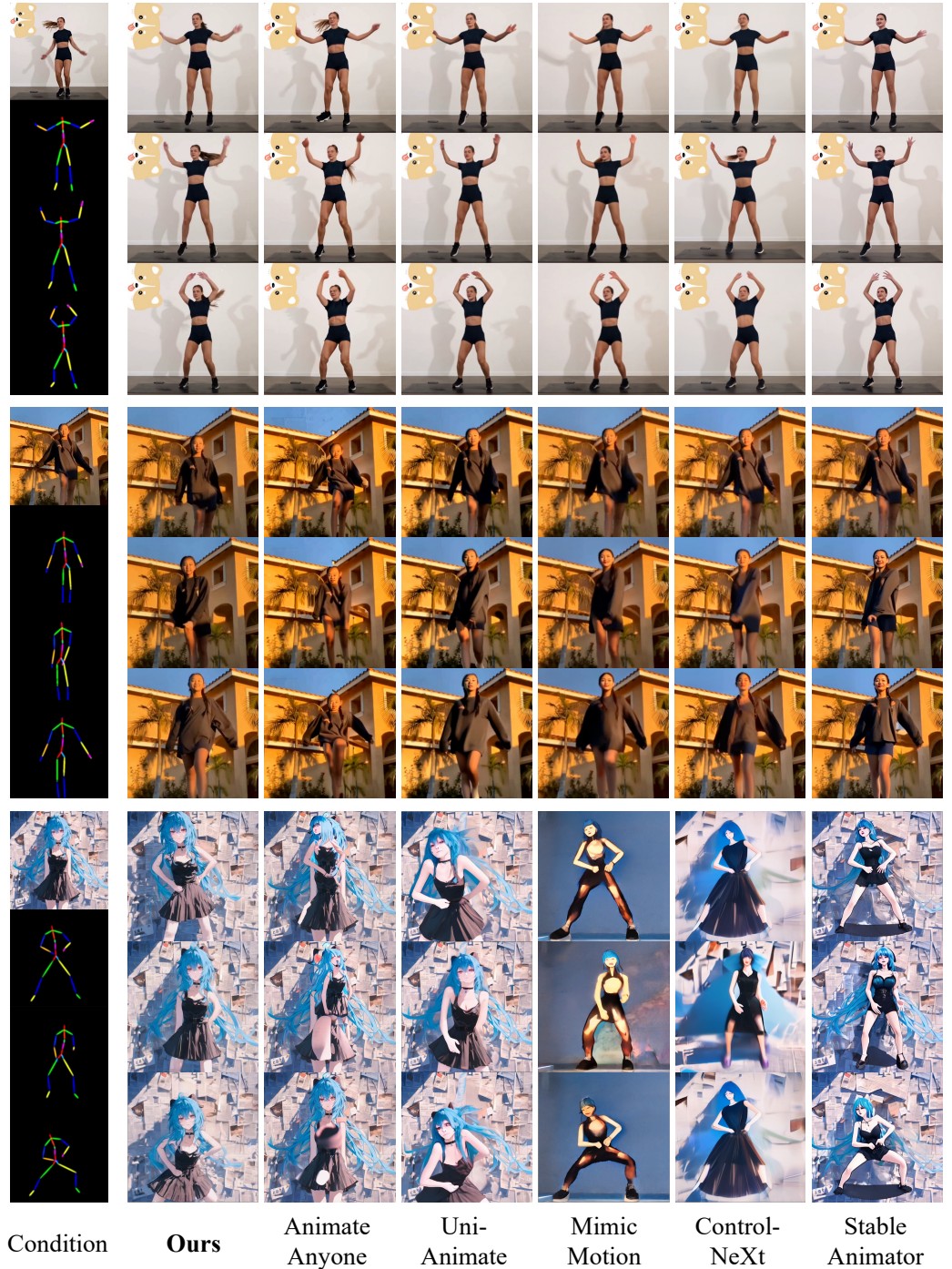

| Condition | **Ours** | Animate Anyone | Uni-Animate | Mimic Motion | Control-NeXt | Stable Animator |

Figure 15: **More Comparisons (2).** Our MTVCraft consistently demonstrates the best performance with high-quality human motion and high-fidelity appearance across different styles and scenes.

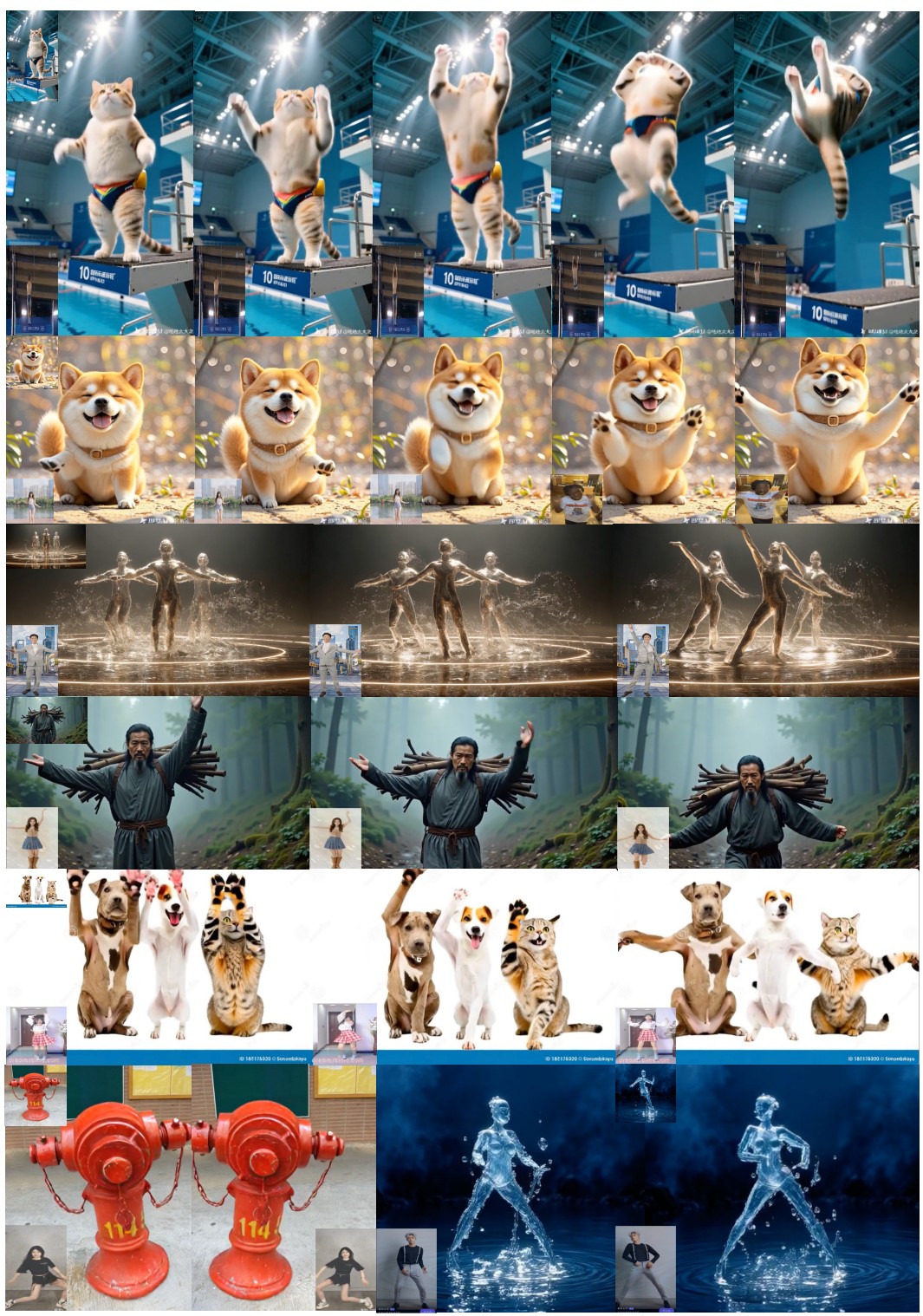

Figure 16: **More Visualization Results (2).** Each row shows an animation conditioned on a different motion sequence. These visualizations showcase our strong zero-shot generalization capability to complex motions, diverse unseen subjects, including animals and even objects.

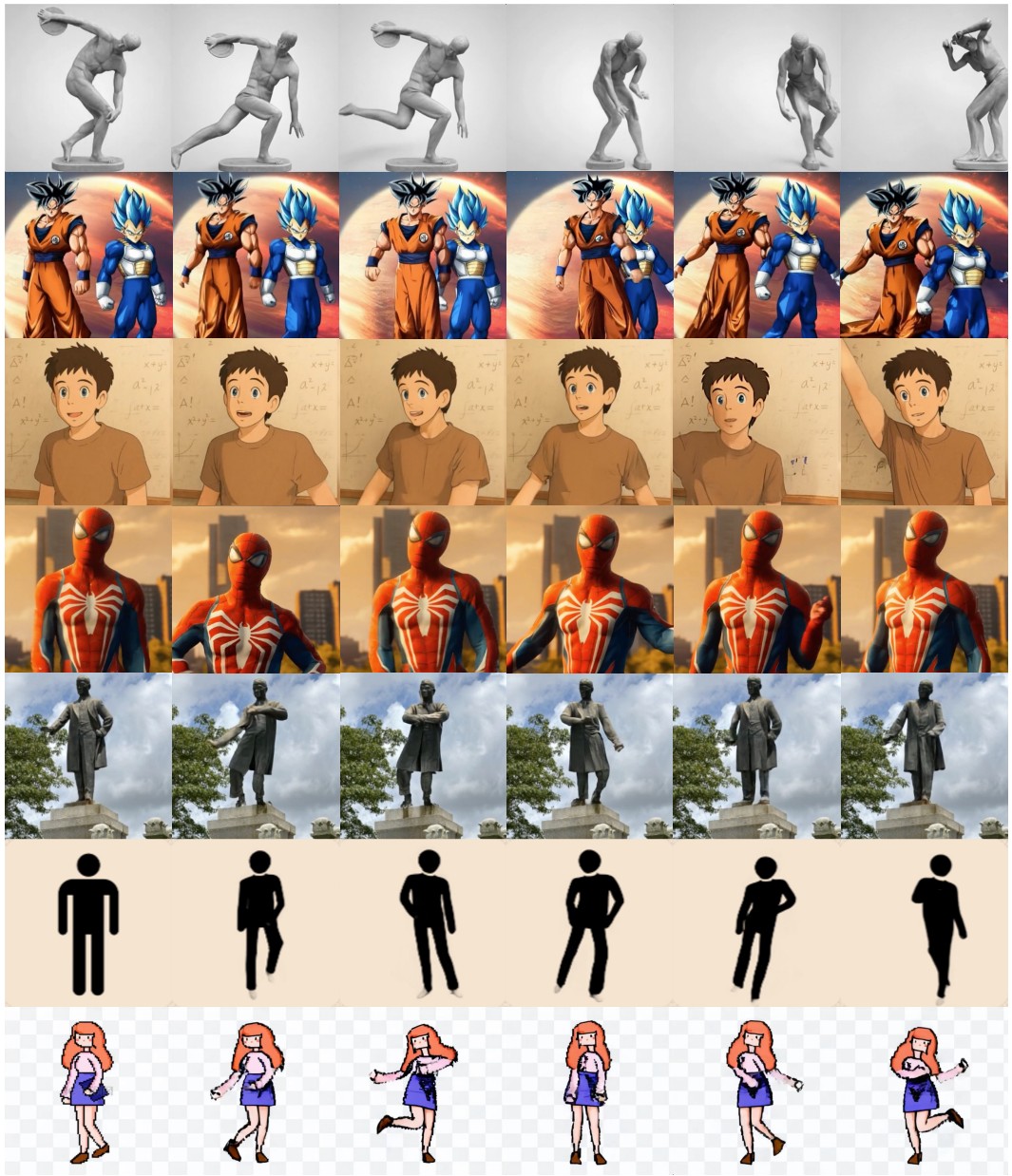

Figure 17: **More Visualization Results (3).** Each row shows an open-world character animation conditioned on a different motion sequence. These visualizations showcase open-world animation results featuring virtual human characters. MTVCraft consistently achieves high identity consistency and motion accuracy across various styles.

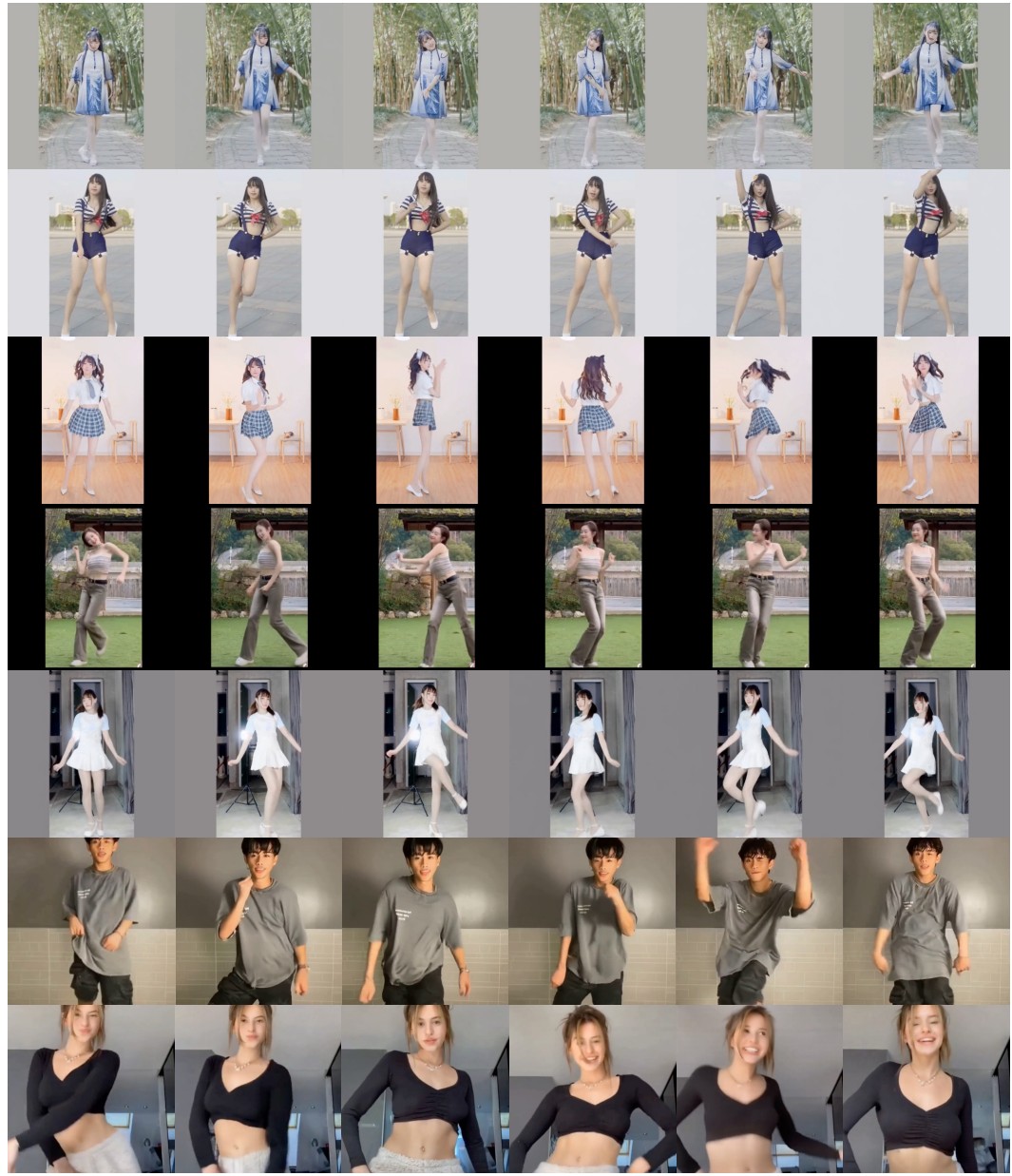

Figure 18: **More Visualization Results (4).** Each row shows a human character animation conditioned on a different motion sequence. Our MTVCraft consistently preserves both identity and motion accuracy across a wide variety of scenarios and diverse real human characters.

