# OpenReview forum: "MTVCraft: Tokenizing 4D Motion for Arbitrary Character Animation"
_ICLR.cc/2026/Conference — ICLR 2026 Poster_

### Official Review · Reviewer_1LKS · 2025-10-21

**Soundness:** 3
**Presentation:** 3
**Contribution:** 3
**Rating:** 8
**Confidence:** 3

**Summary:**

This paper introduces a new framework that directly models raw 3D motion sequences to animate arbitrary characters from reference images. Unlike previous methods that rely on 2D pose renderings, MTVCraft encodes 3D joint trajectories into compact 4D motion tokens using a 4D Motion Tokenizer (4DMoT), which preserves spatial-temporal dynamics and eliminates the need for pixel-level pose alignment. These tokens are then integrated into a Motion-aware Video Diffusion Transformer (MV-DiT) featuring 4D motion attention and 4D positional encodings, enabling expressive, disentangled control of motion and appearance. Implemented on both CogVideoX-5B and Wan-2.1-14B backbones, MTVCraft achieves state-of-the-art results, with superior zero-shot generalization to unseen characters, styles, and non-human subjects.

**Strengths:**

A key strength of integrating a motion-generation pipeline into video generation lies in its ability to provide explicit temporal and structural control over motion, resulting in more coherent and realistic dynamics than end-to-end pixel-based video synthesis. By introducing intermediate motion representations, such as in motion generation domain [1,2], the framework captures fine-grained spatial-temporal cues that general video models often overlook. This separation of motion from appearance allows the generator to synthesize consistent, expressive, and physically plausible movements while preserving identity and style, effectively bridging human motion understanding with visual generation. Moreover, as demonstrated in previous literature [3], coupling motion modeling within video diffusion enables strong generalization to arbitrary characters. Also due to the modular paradigm, the method scales naturally across different backbones, whether smaller transformer-based models like CogVideoX or Wan-2.1, since the motion tokens are from unified SMPL.

[1] Momask : MoMask: Generative Masked Modeling of 3D Human Motions, cvpr 2024
[2] SALAD : salad skeleton-aware latent diffusion for text-driven motion generation and editing, cvpr 2025
[3] AnyMoLe : AnyMoLe: Any Character Motion In-betweening Leveraging Video Diffusion Models, cvpr 2025

**Weaknesses:**

A potential weakness of this paradigm is that the 4D motion compression via 4DMoT is conceptually straightforward and not architecturally novel. The encoder-decoder with vector quantization closely follows standard VQVAE formulations, and while it effectively transforms SMPL joint trajectories into compact motion tokens, it does not introduce fundamentally new techniques in motion encoding or representation learning. However, despite this structural simplicity, the usage and integration of such motion tokenization within a large-scale video generation framework remains a meaningful contribution.

**Questions:**

The codebook size of 8,192 with a 3072-dimensional embedding is relatively large compared to those commonly used in motion generation models. Would decreasing its dimension or size help reduce unused codes or improve resolution?

---

> ### Author Response · Authors · 2025-11-14
> **Reply to Reviewer 1LKS**
>
> ### **Weakness 1**
> *Conceptual Simplicity of 4DMoT*
> ### **Answer to Weakness 1**
> Thank you for your valuable observation.
>
> While the 4DMoT encoder-decoder follows a standard VQVAE formulation, its contribution lies in **effectively applying 4D motion tokenization for character image animation**. We successfully integrate 4D motion tokens into a large-scale controllable video generation framework, providing **novel motion guidance** for this task.
>
> Moreover, as discussed in "Tokenization Strategy: Coordinates vs. SMPL Parameters" (**Lines 295–306**), we use **joint coordinates** as motion input rather than traditional SMPL parameters commonly used in motion generation [1-4]. This constitutes **a novel motion input format**, combined with **different dilated 2D convolutional layers along the frames and joints axes**.
>
> **In summary**, our design of 4DMoT reflects careful considerations. The novelty of 4DMoT lies **not** in the VQVAE architecture itself, **but** in its effective integration for 4D motion-controllable generation and its novel input format, enabling state-of-the-art performance in character image animation.
>
> ---
>
> ### **Question 1**
> *Would decreasing code dimension or codebook size help reduce unused codes or improve resolution?*
> ### **Answer to Question 1**
> Thank you for your insightful question.
>
> We use a codebook of 8,192 entries with 3,072-dimensional embeddings to **balance representation capacity and motion fidelity**.
> - The 3,072-dimensional embeddings allow the codebook to capture rich spatiotemporal correlations, which are **critical** for high-quality motion-controllable generation. They also **align directly with DiT's token dimension**, avoiding additional transformation.
> - As shown in **Figure 8**, nearly 70% of the codes remain active during inference, indicating efficient utilization of the encoding space. Moreover, the cosine similarity of most code pairs is close to zero, confirming a discrete latent space with highly decorrelated representations.
> - Previous work, such as BAD [2], also uses a codebook of 8,192 entries; however, since their task is motion generation, their embedding dimension is smaller than 3,072.
>
> **Regarding decreasing the codebook size or embedding dimension:**
>
> Reducing the codebook size might increase code reuse but also risk degrading motion fidelity. Lowering the embedding dimension could limit the capacity to encode rich spatiotemporal patterns and also break the native alignment with DiT’s token dimension. **Hence, smaller codebooks or embeddings might reduce unused codes slightly, but could also impact motion detail and overall video generation quality.** We agree that it could be an interesting direction for future exploration to investigate trade-offs between code utilization and motion fidelity.
>
> **In summary**, our codebook ensures rich spatiotemporal representation, efficient code utilization, and direct alignment with DiT tokens, which together enable high-quality 4D motion-controllable video generation.
>
> ---
>
> ### References
>
> [1] MoMask: Generative Masked Modeling of 3D Human Motions
>
> [2] BAD: Bidirectional Auto-regressive Diffusion for Text-to-Motion Generation
>
> [3] MotionGPT: Human Motion as a Foreign Language
>
> [4] T2M-GPT: Generating Human Motion from Textual Descriptions with Discrete Representations

---

> > ### Comment · Reviewer_1LKS · 2025-11-26
> >
> > Thank you for the clear rebuttal. I understand the authors’ point that although the 4DMoT architecture follows a standard VQVAE structure, its novelty comes from how motion tokens are designed, using joint coordinates instead of SMPL parameters, and how they are integrated into a large controllable video generation framework. The explanation about the codebook size and embedding dimension make sense, especially given the high code utilization and alignment with DiT token dimensions. While there is still room for future exploration in more compact tokenization and specialized architectural design, this does not weaken the current contribution. Overall, the rebuttal adequately addresses my concerns, and I will maintain my accept recommendation.

---

### Official Review · Reviewer_M8dQ · 2025-10-28

**Soundness:** 3
**Presentation:** 3
**Contribution:** 2
**Rating:** 6
**Confidence:** 4

**Summary:**

This paper proposes MTVCraft (Motion Tokenization Video Crafter), a novel framework for character image animation that directly models raw 3D motion sequences (referred to as 4D motion) rather than relying on traditional 2D-rendered pose images. This is achieved by introducing a 4D Motion Tokenizer (4DMoT) to quantize 3D motion into compact tokens, which are then used to condition a Motion-aware Video DiT (MV-DiT). The approach effectively decouples motion from pixel-level alignment and appearance biases, showing strong zero-shot generalization to arbitrary characters, diverse styles, and non-human subjects. MTVCraft achieves state-of-the-art performance on benchmarks like TikTok and Fashion. The shift from 2D pose to 4D motion tokens is a significant and positive step forward for controllable video generation.

**Strengths:**

1.	The paper addresses a limitation of current methods by replacing fragile 2D pose images with robust, compact 4D motion tokens derived directly from SMPL joint coordinates.
2.	Experimental results indicate the effectiveness of the proposed method.

**Weaknesses:**

1.	The method introduces a new, independently trained component: the 4D Motion Tokenizer (4DMoT). Training this additional encoder (a VQVAE) adds complexity to the overall pipeline and represents an extra component that must be learned, stored, and maintained, potentially limiting the ability to scale the entire framework compared to methods that use off-the-shelf 2D pose estimators. During inference, the system requires an additional forward pass through the 4DMoT encoder to generate the motion tokens from the raw SMPL joint coordinates. While the tokens are compact, the initial encoding step introduces additional inference latency and computational cost that is not present in 2D-based methods (which often use pre-calculated 2D maps). The paper should provide a detailed breakdown of the latency and resource consumption of the 4DMoT during inference compared to the latency of the main MV-DiT model to justify this extra computational step.
2.	The entire pipeline is contingent upon the accuracy of the upstream SMPL joint sequence estimation (using NLF-Pose in this work ). Errors or noise in the initial 3D pose data will directly impact the quality of the 4D motion tokens and thus the final animation quality.
3.	The teaser figure shows multi-character animation results. But the motions of different characters are same. How can we animate only one character in the multi-character image?

**Questions:**

Why 3D/4D information is more useful than 2D? Can the authors provide some evidence?

---

> ### Author Response · Authors · 2025-11-14
> **Reply to Reviewer M8dQ (Part 1/2)**
>
> ### **Weakness 1**
> *Additional Complexity of 4DMoT*
> ### **Answer to Weakness 1**
> Thank you for your valuable comment.
>
> While 4DMoT introduces an additional VQVAE [1], it is **lightweight, cost-efficient, and highly extensible**, adding minimal complexity to the overall pipeline.
>
> - **Model size**: 4DMoT contains only **48M** parameters (**0.8%** of the 6B backbone and **0.3%** of the 18B backbone), making its training cost negligible relative to the backbone.
> - **Inference cost**: As shown in the table below, encoding 49 frames of SMPL joints (49×24×3 inputs) into motion tokens takes **0.21 seconds** on a single A100 GPU, which is less than **1%** of the total inference time.
> - **Scalability and maintainability**: Our 4DMoT is **modular and easily adaptable to different backbones**. For example, we successfully implemented it on both CogVideo-X [2] and Wan-2-1 [3] **without any retraining**. It also demonstrates **strong generalization** on unseen gymnastics data, as shown in **Figure 9**.
> - **Compared with 2D-based methods**: 2D rendered pose images usually require a pixel-level pose encoder to extract conditional features. This process becomes increasingly expensive at high resolutions because 2D-based encoders are **resolution-dependent**. In contrast, our 4DMoT is **resolution-free** (input size is always frame_num × 24 × 3).
>
> |**Model**|**Pose Encoder Size**|**Input Motion Size**|**Pose Encoder Latency**|**Total Inference Latency**|
> |-|-|-|-|-|
> |Unianimate-DiT [4]|0.38M|49×(512×512)×3|0.06s|197s|
> |Unianimate-DiT [4]|0.38M|49×(1024×1024)×3|0.27s|1061s|
> |**MTVCraft-6B**|**48M**|**49×24×3**|**0.21s**|**87s**|
> |**MTVCraft-18B**|**48M**|**49×24×3**|**0.21s**|**153s**|
>
> As resolution increases, the pose encoder latency of UniAnimate-DiT grows, whereas 4DMoT’s latency remains constant. This makes 4DMoT **more efficient in higher-resolution settings**. Overall, pose encoding latencies are **negligible** for both methods, while 4DMoT brings **substantial performance gains** over traditional pixel-level pose encoders, as demonstrated in our quantitative and qualitative results **(Tables 1, 3; Figures 1, 5, 13, 15)**.
>
> ---
>
> ### **Weakness 2**
> *SMPL Joint Estimation Accuracy*
> ### **Answer to Weakness 2**
> Thank you for pointing this out.
>
> While our work does not aim to improve SMPL estimation itself, we **carefully curated and filtered the training data** to mitigate potential inaccuracies, as detailed in **Appendix B**.
>
> 1. **Confidence-based filtering**: We compute per-sequence pose uncertainty from per-joint scores and retain only sequences below 0.9× the global dataset average.
> 2. **Motion magnitude filtering**: Optical flow [5] is used to remove sequences with extreme motion, reducing the difficulty of pose estimation. The threshold is set to 2.
> 3. **Manual inspection**: 200 randomly sampled clips were checked, consistently showing accurate motion estimations. Projected back to 2D, the 3D poses align well with 2D RTMPose [6] detections, whose accuracy under normal motions is sufficient for verification.
>
> In more challenging or high-speed actions, 3D mesh estimation is often more stable and preserves full-body structure.  As shown in **Figures 9, 13, and 16**, SMPL poses are accurately estimated even in complex gymnastics sequences. Our pipeline **generalizes well to these complex motions outside the training set**, suggesting that the training data is effective and the 4DMoT is robust. Moreover, compared with SMPL-rendering-based approaches (e.g., Realisdance-DiT [7]), ours performs better on both benchmarks (**Tables 1, 3**).
>
> ---
>
> ### References
>
> [1] Neural Discrete Representation Learning
>
> [2] CogVideoX: Text-to-Video Diffusion Models with An Expert Transformer
>
> [3] Wan: Open and Advanced Large-Scale Video Generative Models
>
> [4] UniAnimate-DiT: Human Image Animation with Large-Scale Video Diffusion Transformer
>
> [5] Unifying Flow, Stereo and Depth Estimation
>
> [6] Rtmpose: Real-time multi-person pose estimation based on mmpose
>
> [7] RealisDance-DiT: Simple yet Strong Baseline towards Controllable Character Animation in the Wild

---

> ### Author Response · Authors · 2025-11-14
> **Reply to Reviewer M8dQ (Part 2/2)**
>
> ### **Weakness 3**
> *Animating Only One Character in Multi-Character Scenes*
> ### **Answer to Weakness 3**
> Thank you for your insightful question.
>
> The teaser shows **consistent motion transfer across multiple characters**, but this is a visualization choice, not a limitation. The framework naturally supports **per-character control**, with two extensions:
>
> 1. **Non-interacting multiple characters:** One can animate a specific character by segmenting it (e.g., using SAM [7]) and applying 4DMoT tokens only to the masked region, treating other characters as static. This works when characters do not strongly overlap.
>
> 2. **Interacting or overlapping characters:** For complex multi-character scenes, one can bind different motion-token sequences to different character IDs along the token dimension, allowing simultaneous and independent motion control while preserving interactions.
>
> Thus, our method provides **a general and extensible foundation for multi-character asynchronous motion control**.
>
> ---
>
> ### **Question 1**
> *Why 3D/4D information is more useful than 2D?*
> ### **Answer to Question 1**
> Thank you for your valuable question.
>
> Our 4D motion tokens provide more powerful and richer guidance than 2D-based methods, as illustrated below:
>
> 1.	**Rich spatiotemporal motion representation:**
>
> 2D images provide only pixel-level cues, which **cannot fully describe complex motions in real 4D scenes**. For example, in **Figure 13**, gymnastics sequences involve rotations and flips that 2D-based methods fail to handle. In contrast, our 4D motion tokens encode both temporal dynamics and 3D spatial structure, enabling accurate modeling of such motions.
>
> 2.	**Depth and scale ambiguity**
>
> 2D-only methods **cannot directly infer depth or relative scale**, which may cause ambiguous motion interpretation (e.g., a small person in the image may be far from the camera or actually small). Our 4D motion tokens naturally **resolve such ambiguities**, providing consistent motion representation across different viewpoints and object sizes.
>
> 3. **Empirical evidence**
>
> Tables 1 and 3 quantitatively demonstrate our state-of-the-art results. Ablation studies in Tables 2 and 4 further show that **quantizing 4D (txyz) outperforms 3D-only (txy) quantization**. Figures 1, 2, 5, 13, and 15 qualitatively illustrate the advantages of our 4D motion tokens over traditional 2D-based methods.
>
> **In summary**, our 4D motion tokens provide richer motion and depth cues, enabling accurate and robust motion modeling beyond the capabilities of 2D-based methods.
>
> ---
>
> ### References
>
> [1] SAM 2: Segment Anything in Images and Videos

---

### Official Review · Reviewer_KU6U · 2025-10-28

**Soundness:** 4
**Presentation:** 3
**Contribution:** 4
**Rating:** 8
**Confidence:** 4

**Summary:**

This paper proposes MTVCraft, which shifts pose guidance from 2D renderings to direct tokenization of raw 4D motion (SMPL joint coordinates) via a VQ-VAE motion tokenizer (4DMoT). These motion tokens condition a Motion-aware Video DiT (MV-DiT) equipped with 4D motion attention and 4D rotary positional encoding, aligning spatio-temporal structure during generation. The system scales from a 6B (CogVideoX-5B) to an 18B backbone (Wan-2.1-14B) and supports multi-control (text + motion) with simple integration, delivering state-of-the-art results on TikTok and Fashion benchmarks. Design ablations and negative results further motivate coordinate-space tokenization over parameter-space alternatives.

**Strengths:**

1. Clear paradigm shift from 2D renderings to discrete 4D motion tokens, with a well-articulated rationale (coordinates vs. parameters) and an architecture that uses motion tokens natively (4D RoPE + motion attention).

2. Strong empirical gains on both TikTok and Fashion. The 18B model improves FID/FVD while modestly raising SSIM/PSNR over strong baselines.

3. Scalable & practical. The 18B integration is straightforward (zero-padding alignment), and the paper documents unsuccessful alternatives (linear/MLP projection, SMPL-parameter tokenizer), which is valuable for reproducibility and future work.

4. This paper shows cross-identity animation results across different species, e.g., human pose to bird, fish or even chair.

**Weaknesses:**

1. Camera view handling are implicit. There’s no explicit camera-parameter conditioning. The method relies on data diversity and 4D tokens. This is workable, but leaves questions about view transitions or long-term 3D consistency.

2. As SMPL joint are hard to accurately estimated, how the authors ensure that annotation quality? Besides, why don't use SMPL-X which includes hands?

3. For the heavy 18B model, will the inference cost of the proposed model be 10x or even 100x of the previous U-net based models, so that the comparison is not fair enough? It will be better to report the inference cost in performance comparison.

**Questions:**

Please see the weaknesses.

Typos: "Fahsion" should be "Fashion" in Table.4's header.

Overall, this paper offers a clean, well-justified shift from 2D renderings to discrete 4D motion tokens, demonstrates superior results on two standard benchmarks (with scaling to 18B), and provides enough architectural detail to be useful for practitioners. The main remaining gaps ( camera parameterization, brief data quality discussion) are addressable in the camera-ready and do not undermine the core contribution. I believe this paper will contribute to the community, so I vote clear acceptance.

---

> ### Author Response · Authors · 2025-11-14
> **Reply to Reviewer KU6U**
>
> ### **Weakness 1**
> *Implicit Camera View Handling*
> ### **Answer to Weakness 1**
> Thank you for pointing this out.
>
> Our method leverages the NLF pose estimator [1], which assumes **a default camera with a 55° field of view and a centered principal point**. All predicted 3D joints are represented in the absolute camera coordinate system with consistent intrinsic parameters. Moreover, our training data is captured under **fixed or near-fixed camera settings**, contributing to more reliable pose estimation.
>
> This design simplifies view ambiguity and allows the model to focus on **learning precise human pose dynamics rather than camera motion**. At the same time, the 4DMoT tokenizer provides a **temporally coherent 4D motion representation**, which helps maintain long-term 3D consistency.
>
> Extending to full camera control is feasible and could be explored in future work. For instance, we can add camera-view attention after the motion attention and include an additional training stage dedicated to camera-view conditioning, where the camera tokens can be Plücker embeddings [2].
>
> ---
>
> ### **Weakness 2**
> *SMPL Joint Estimation Accuracy and SMPL-X Choice*
> ### **Answer to Weakness 2**
> Thank you for the valuable comment.
>
> While our work does not aim to improve SMPL estimation itself, we **carefully curated the training data to mitigate potential inaccuracies (Appendix B)**. Specifically, we **filter poses** using confidence scores [1] and motion magnitude [3]. We also **manually inspected** 200 randomly sampled clips and consistently observed accurate motion estimates. When projected back to 2D, the estimated 3D poses align well with RTMPose [4] detections, whose accuracy under normal motions is sufficient for verification.
>
> In more challenging or high-speed actions, 3D mesh estimation is often more stable and preserves full-body structure. Qualitative examples in **Figures 9, 13, and 16** further demonstrate accurate reconstruction on demanding gymnastics data, confirming the effectiveness of both our data and motion tokenizer. Finally, compared with SMPL-rendering-based approaches (e.g., RealisDance-DiT [9]), our method achieves superior performance on both benchmarks in Tables 1 and 3.
>
> We chose **not** to use SMPL-X for the following reasons:
>
> - **SMPL-X estimation is more challenging and thus less reliable**, especially for capturing subtle hand articulations. It remains a major challenge in pose estimation.
>
> - **Training with SMPL-X tokenizers demands videos with significantly higher hand resolution and close-up views**, which are difficult to collect at scale.
>
> Nonetheless, our framework is fully extensible to SMPL-X, and once high-quality hand data are available, the motion tokenizer can easily incorporate hand motions. In short, SMPL provides a simple and effective solution for current full-body pose control, while extension to SMPL-X is feasible for future work.
>
> ---
>
> ### **Weakness 3**
> *Inference Cost of Large 18B Model*
> ### **Answer to Weakness 3**
> We tested different models on 49 frames at a resolution of 512×512 using a single A100 GPU (25 inference steps):
>
> |**Model**|**Inference Cost**|**Model Size**|
> |-|-|-|
> |MimicMotion [5]|42s|1.5B|
> |Animate-X [6]|78s|1.8B|
> |Unianimate-DiT [7]|197s|16B|
> |**MTVCraft-6B**|**87s**|**6B**|
> |**MTVCraft-18B**|**153s**|**18B**|
>
> Compared with smaller U-Net–based [8] models, our framework incurs about **2×** higher inference cost but achieves significantly improved controllability and generation quality. Although the parameter count increases by 10×, the latency grows much more slowly because DiT-based architectures scale more efficiently in parallel computation. This trade-off is **acceptable** given the **significant performance gains** in both quantitative and qualitative results.
>
> ---
>
> ### **Question 1**
> *Typos: Fahsion*
> ### **Answer to Question 1**
> Thank you for pointing out that “Fahsion” should be corrected to “Fashion” in Table 4’s header. We will fix this typo in the camera-ready version.
>
> ---
>
> ### References
>
> [1] Neural localizer fields for continuous 3d human pose and shape estimation
>
> [2] Light Field Networks: Neural Scene Representations with Single-Evaluation Rendering
>
> [3] Unifying Flow, Stereo and Depth Estimation
>
> [4] Rtmpose: Real-time multi-person pose estimation based on mmpose
>
> [5] MimicMotion: High-Quality Human Motion Video Generation with Confidence-aware Pose Guidance
>
> [6] Animate-X: Universal Character Image Animation with Enhanced Motion Representation
>
> [7] UniAnimate-DiT: Human Image Animation with Large-Scale Video Diffusion Transformer
>
> [8] U-Net: Convolutional Networks for Biomedical Image Segmentation
>
> [9] RealisDance-DiT: Simple yet Strong Baseline towards Controllable Character Animation in the Wild

---

### Public Comment · ~Shuolin_Xu1 · 2025-11-12
**About 4D token**

very Solid work

---

> ### Author Response · Authors · 2025-11-14
> **Reply to Shuolin Xu**
>
> Thank you very much for your kind words! We plan to open-source the code and checkpoints for the community.

---

### Author Response · Authors · 2025-11-14
**Global Reply**

### Dear Reviewers,

We sincerely thank all reviewers for their thoughtful comments and constructive feedback.

All reviewers acknowledge the strengths of our work, noting that it is well motivated, practical, and demonstrates strong empirical performance, with scores of 8, 6, and 8 respectively. Reviewers also raised several important questions, such as the accuracy of SMPL estimation and the inference latency.

We have carefully considered each point to clarify, justify, and discuss potential future improvements. Detailed responses are provided below. We hope our explanations fully resolve the reviewers’ concerns. Please do not hesitate to let us know if there are any additional details or clarifications that would be helpful.

### Sincerely,
### MTVCraft Authors

---

### Meta-Review · Area_Chair_WjWb · 2026-01-15

**Summary:**

The paper puts forward a relatively simple idea that seems to be working on the one benchmark, where quantitative evaluation is performed. The idea is to extend motion tokenizers so that they world on 4D, after extracting 3D sillhouettes from video using SMPL. The architecture is novel in a conventional way: previous methods focus on 2D variants, while this one extends the modelling paradigm to 4D, using off-the-shelf components and methodologies.

The reviews are positive; however, they are not critical enough of the single most important aspect for a scientific paper: novelty.  I would say that the paper is on the low end, also evidenced by the description of the method, which primarily concerns discussing implementation details and how existing methodologies are reused.

On the evaluation, the method could be more thorough. The main paper reports quantitative results only on a single dataset (TikTok), while a second benchmark is used for visual inspection. Results for Fashion are also reported in the appendix, where improvements are consistent but small (in terms of PSNR and SSIM). Importantly, almost all compared methods are arXiv papers, which means that the evaluation is also not very convincing. Also, the evaluations are between methods of different complexity, so they reflect 'state-of-the-art' rather than fair "apple-to-apple" comparisons.

Overall, I would say that while the reviews are positive, the scope of the paper is not sufficient to convincingly defend its novel contribution for a conference like ICLR, which focuses on learning representations, not visual generation. That said, given that the reviews are unanimously positive even if they could be more critical, I recommend acceptance.

**Reviewer Concerns:**

The limited novelty was the most important comment raised by one reviewer.

Other concerns included engineering aspects of the work such as the inference cost and latency, as well hyperparameter settings (camera position, choice of baseline implementations like SMPL vs SMLP-X).

**Reviewer Scores:**

All reviewers are in fact positive (2x 8, 1x 6).

I cannot really answer this, but I think the reviewers could have been more critical (in a fair and positive way), assuming being prompted from the AC. From where I stand, there is certainly something new, but I am not convinced that what is new is relevant to ICLR.

---

### Decision · Program_Chairs · 2026-01-26

Accept (Poster)